# A polarity pathway for exocyst-dependent intracellular tube extension

Joshua Abrams[1], Jeremy Nance[1,2]*

[1]Skirball Institute of Biomolecular Medicine, NYU Grossman School of Medicine, New York, United States; [2]Department of Cell Biology, NYU Grossman School of Medicine, New York, United States

**Abstract** Lumen extension in intracellular tubes can occur when vesicles fuse with an invading apical membrane. Within the *Caenorhabditis elegans* excretory cell, which forms an intracellular tube, the exocyst vesicle-tethering complex is enriched at the lumenal membrane and is required for its outgrowth, suggesting that exocyst-targeted vesicles extend the lumen. Here, we identify a pathway that promotes intracellular tube extension by enriching the exocyst at the lumenal membrane. We show that PAR-6 and PKC-3/aPKC concentrate at the lumenal membrane and promote lumen extension. Using acute protein depletion, we find that PAR-6 is required for exocyst membrane recruitment, whereas PAR-3, which can recruit the exocyst in mammals, appears dispensable for exocyst localization and lumen extension. Finally, we show that CDC-42 and RhoGEF EXC-5/FGD regulate lumen extension by recruiting PAR-6 and PKC-3 to the lumenal membrane. Our findings reveal a pathway that connects CDC-42, PAR proteins, and the exocyst to extend intracellular tubes.

**\*For correspondence:**
jeremy.nance@med.nyu.edu

**Competing interests:** The authors declare that no competing interests exist.

## Introduction

Most organs contain tubes, which are used to transport gases and fluids from one site within the body to another. The circumference of larger tubes, such as the human intestine, is lined by many cells connected to one another with junctions. By contrast, the smallest tubes have intracellular lumens that are contained entirely within the cytoplasm of a cell. Although some intracellular tubes arise when a cell wraps circumferentially and recontacts itself to hollow out a lumen from the extracellular space (*Rasmussen et al., 2008*; *Stone et al., 2009*), many intracellular tubes are thought to form when an apical membrane domain invades into the cytoplasm to become the lumen (*Lubarsky and Krasnow, 2003*; *Sundaram and Cohen, 2017*). The *Caenorhabditis elegans* excretory cell provides a powerful model system for studying this mechanism of intracellular lumen extension. Born during the first half of embryogenesis, the H-shaped excretory cell contains four long canal arms that grow during larval stages to extend nearly the full length of the worm by the beginning of the L2 larval stage (*Nelson et al., 1983*; *Sundaram and Buechner, 2016*). An intracellular lumen initiates within the cell body and invades the length of each canal arm, functioning in osmoregulation (*Buechner et al., 1999*; *Mancuso et al., 2012*; *Nelson and Riddle, 1984*; *Sundaram and Buechner, 2016*). Vertebrate capillaries, as well as terminal and fusion cells of the *Drosophila* trachea and the *Ciona* notochord, are additional examples of cells containing intracellular tubes that are thought to form through an apical invasion mechanism (*Denker et al., 2013*; *Gervais and Casanova, 2010*; *Herwig et al., 2011*; *Lenard et al., 2013*).

Extension of an intracellular lumen by apical domain invasion requires the polarized delivery and fusion of vesicles, which supply the new membrane needed to expand the lumenal surface (*Berry et al., 2003*; *Gervais and Casanova, 2010*; *Khan et al., 2013*; *Kolotuev et al., 2013*; *Schottenfeld-Roames and Ghabrial, 2012*). The highly conserved, eight-protein exocyst complex and the small GTPase exocyst activator Ral are required for polarized membrane targeting of vesicles in

many cell types (*Wu and Guo, 2015*). The exocyst mediates vesicle tethering and subsequent fusion at sites where it enriches on the cell membrane (*He and Guo, 2009*; *Lipschutz et al., 2000*; *Liu and Guo, 2012*). Studies in both yeast and mammalian cells suggest that the eight exocyst subunits (Sec3, Sec5, Sec6, Sec8, Sec10, Sec15, Exo70, and Exo84) assemble together from distinct subcomplexes to promote vesicle tethering (*Ahmed et al., 2018*; *Heider et al., 2016*). Active Ral GTPase binds directly to the exocyst to promote its assembly (*Brymora et al., 2001*; *Chen et al., 2011*; *Moskalenko et al., 2002*; *Moskalenko et al., 2003*; *Sugihara et al., 2002*). The exocyst is enriched at the lumenal membrane of *Drosophila* and *C. elegans* intracellular tubes and is required for lumen extension (*Armenti et al., 2014a*; *Jones et al., 2014*), suggesting that it targets the vesicles needed for membrane expansion. A key unanswered question is how exocyst localization becomes polarized to accumulate on the lumenal membrane.

PAR proteins, which include Par3 (a multi-PDZ domain scaffolding protein), Par6 (a PDZ and CRIB domain scaffolding protein), and aPKC (atypical protein kinase C), mediate cell polarity by establishing an asymmetric signaling domain at the plasma membrane (*Nance and Zallen, 2011*; *St Johnston and Ahringer, 2010*). Upstream polarity cues can induce PAR asymmetries by activating the Rho GTPase Cdc42, which binds directly to the Par6 CRIB domain, recruiting Par6 and its binding partner aPKC to the membrane and promoting aPKC kinase activity (*Aceto et al., 2006*; *Gotta et al., 2001*; *Hutterer et al., 2004*; *Joberty et al., 2000*; *Johansson et al., 2000*; *Kay and Hunter, 2001*; *Lin et al., 2000*; *Qiu et al., 2000*). Par6 and aPKC are also concentrated within asymmetric membrane domains by interacting with Par3 (*Tabuse et al., 1998*; *Watts et al., 1996*). PAR proteins regulate downstream effectors through aPKC phosphorylation or by recruiting effector proteins directly (*Nance and Zallen, 2011*; *St Johnston and Ahringer, 2010*).

PAR proteins are important for lumen expansion in both multicellular and intracellular tubes. For example, in MDCK multicellular cysts grown in 3D culture, Par3 localizes to the membrane of the lumen that forms at the center of the cell cyst, and its knockdown leads to the formation of multiple, disorganized lumens (*Bryant et al., 2010*). In *Drosophila* terminal tracheal cells, Par-6 and aPKC are found at the lumenal membrane and are thought to be required for lumenogenesis (*Jones and Metzstein, 2011*). Within the *C. elegans* excretory cell, fluorescently tagged PAR-3 and PAR-6 expressed from transgenes, and endogenous PAR-6 and PKC-3/aPKC detected by immunostaining, accumulate at the lumenal membrane (*Armenti et al., 2014a*). Transgenic CDC-42 and a putative activator, the RhoGEF EXC-5/FGD, are also enriched at the lumenal membrane (*Lant et al., 2015*; *Mattingly and Buechner, 2011*; *Suzuki et al., 2001*). Whereas *exc-5* mutants have severely truncated excretory cell canals (*Buechner et al., 1999*; *Gao et al., 2001*; *Suzuki et al., 2001*), the contribution that PAR proteins and CDC-42 make to excretory cell lumen extension has not been fully determined because these proteins have earlier essential developmental functions (*Gotta et al., 2001*; *Kay and Hunter, 2001*; *Kemphues et al., 1988*; *Tabuse et al., 1998*; *Watts et al., 1996*).

Several PAR proteins have been shown to physically interact with the exocyst (*Ahmed and Macara, 2017*; *Das et al., 2014*; *Lalli, 2009*; *Rosse et al., 2009*; *Zuo et al., 2011*; *Zuo et al., 2009*), raising the possibility that PAR proteins might function in lumen extension by recruiting the exocyst to the lumenal membrane. In mammary epithelial cells, a lysine-rich domain of Par3 binds directly to the exocyst protein Exo70 and is thought to function as an exocyst receptor, recruiting the complex to sites where Par3 is enriched (*Ahmed and Macara, 2017*). Within migrating rat kidney epithelial cells, aPKC interacts with the exocyst through the aPKC-binding protein Kibra and is required for exocyst enrichment at the leading edge, although exocyst is also required for aPKC localization to this site (*Rosse et al., 2009*). In mammalian neurons, the PDZ domain of Par6 can bind the exocyst (through Exo84), and this interaction requires active Ral GTPase (*Das et al., 2014*). These observations raise the possibility that Par3, Par6, and/or aPKC are required to enrich the exocyst at the lumenal membrane during intracellular tube extension. Consistent with this model, Sec8 enrichment at the lumenal membrane domain in *aPKC* mutant *Drosophila* terminal tracheal cells is lost (*Jones et al., 2014*). However, the lumen and branching defects of *aPKC* mutant tracheal cells make it difficult to establish whether aPKC recruits the exocyst directly to the lumenal membrane, or whether exocyst loss from the lumenal membrane arises indirectly as a result of other aPKC-dependent cellular defects. Testing whether PAR proteins recruit the exocyst during intracellular tube extension would ideally be accomplished by eliminating PAR proteins acutely, after lumenogenesis is complete, and determining if exocyst localization is altered.

Here, we utilize degron-tagged alleles of SEC-5, RAL-1, PAR-3, PAR-6, PKC-3, CDC-42, and EXC-5 to establish the roles of these proteins in extending the excretory cell intracellular lumen. We show that PAR-6 and PKC-3, but not PAR-3, are essential for lumen extension, and using acute protein depletion we demonstrate that PAR-6, but not PAR-3, is needed to recruit the exocyst to the lumenal membrane. Finally, we provide evidence that EXC-5 and CDC-42 function upstream of PAR-6 and PKC-3 as polarity cues, recruiting these proteins to the lumenal membrane. Our findings identify a pathway that connects Rho GTPase, cell polarity, and vesicle-tethering proteins to lumen extension during intracellular tubulogenesis.

## Results

### SEC-5 and RAL-1 function within the excretory cell to promote lumen extension

The enrichment of the exocyst at the excretory cell lumenal membrane and its requirement for proper lumen extension suggest that exocyst-dependent vesicle delivery provides the new membrane needed for lumen expansion (*Armenti et al., 2014a*). If so, the exocyst, which is broadly expressed and needed for embryonic development (*Armenti et al., 2014a*; *Frische et al., 2007*), should be required autonomously within the excretory cell. To test this hypothesis, we designed a degron-based strategy to conditionally deplete exocyst component SEC-5 and exocyst activator RAL-1 (the sole *C. elegans* Ral GTPase homologue) specifically within the excretory cell (*Figure 1A*); this approach removes zygotically expressed protein as well as inherited maternal protein, which can otherwise mask mutant phenotypes (*Nance and Frøkjær-Jensen, 2019*). Proteins tagged with the ZF1 degron are rapidly degraded to undetectable levels by expressing the E3 ubiquitin ligase substrate-adapter protein ZIF-1 (*Armenti et al., 2014b*; *DeRenzo et al., 2003*; *Reese et al., 2000*). In order to express ZIF-1 specifically within the excretory cell, we searched for an excretory cell-specific promoter. Existing transcriptional reporters for two promoters described to be active predominantly or exclusively in the excretory cell, *pgp-12* (*Zhao et al., 2005*) and *glt-3* (*Mano et al., 2007*), showed additional expression in other embryonic tissues. Using the WormBase (https://wormbase.org/) data-mining platform WormMine, we identified additional candidate promoters among a set of genes described to be expressed specifically within the excretory cell. Upstream sequences of one gene, *T28H11.8,* drove detectable mCherry expression specifically in the excretory cell from embryogenesis onward (*Figure 1—figure supplement 1*), and endogenous *T28H11.8* mRNA is first detected by single-cell RNA sequencing in the excretory cell several hours after its birth (*Packer et al., 2019*). To determine if ZIF-1 expressed from the *T28H11.8* promoter (hereafter *excP*) was sufficient to degrade ZF1-tagged proteins specifically within the excretory cell, we introduced a high-copy array containing *excP::zif-1* into worms expressing a ZF1-tagged reporter protein, ZF1::GFP::CDC-42. Control larvae, which did not inherit the *excP::zif-1* array, robustly expressed ZF1::GFP::CDC-42 in the excretory cell and other tissues (*Figure 1B*). By contrast, ZF1::GFP::CDC-42 was depleted below detectable levels within the excretory cell in larvae that inherited the *excP::zif-1* transgenic array (*Figure 1C*), whereas expression of ZF1::GFP::CDC-42 persisted in other tissues. We conclude that *excP::zif-1* can be used to deplete ZF1-tagged proteins from the excretory cell.

In order to inhibit exocyst activity specifically within the excretory cell, we created a high-copy, integrated *excP::zif-1* transgene to conditionally degrade ZF1-tagged SEC-5 and RAL-1 proteins. For SEC-5, we utilized *sec-5(xn51)*, a functional, endogenously tagged *sec-5::zf1::yfp* allele (*Armenti et al., 2014b*). Similar to SEC-5::YFP protein expressed from a transgene (*Armenti et al., 2014a*), endogenously tagged SEC-5::ZF1::YFP concentrated at the excretory cell lumenal membrane (*Figure 1D*). For RAL-1, we utilized the *ral-1(tm5205)* null mutation rescued by a previously characterized, low-copy, functional *ral-1P::zf1::yfp::ral-1* transgene (*Armenti et al., 2014a*). We examined phenotypes of worms with excretory cell-specific depletion of SEC-5::ZF1::YFP (SEC-5[exc(-)] worms) or ZF1::YFP::RAL-1 (RAL-1[exc(-)] worms) using co-expressed markers of the excretory cell cytoplasm (*excP::yfp*) and lumenal membrane (*ifb-1::cfp*) (see *Figure 1A*). Controls expressing *excP::zif-1* but not the ZF1-tagged proteins displayed normal excretory canal outgrowth and morphology (*Figure 1E,F–F''*). In contrast to controls, SEC-5[exc(-)] and RAL-1[exc(-)] larvae had severely truncated, swollen canals with disorganized, cystic lumens (*Figure 1G–J''*). Small cysts often appeared to be discontinuous, although given the resolution of our imaging, it is possible that they remain

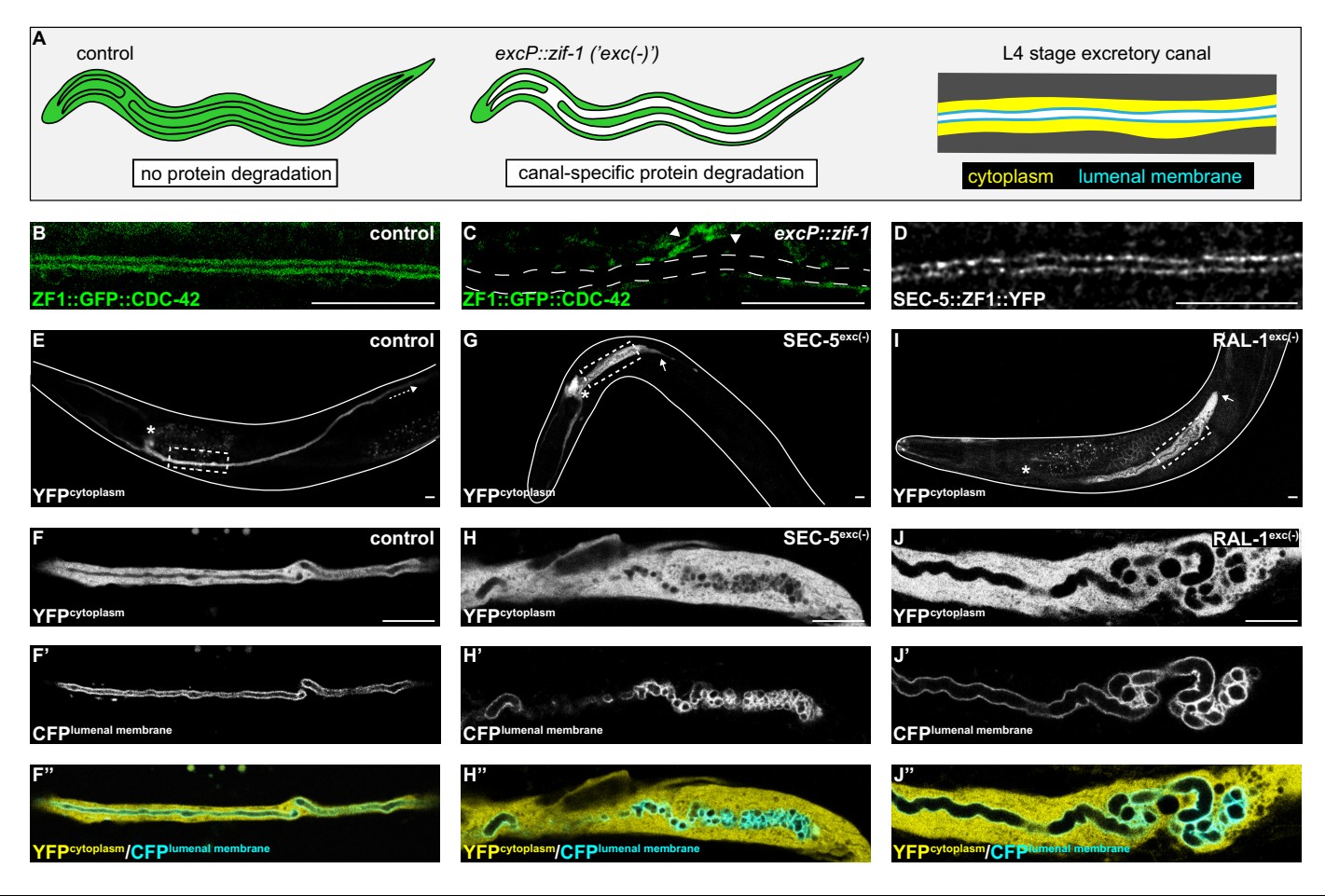

**Figure 1.** SEC-5 and RAL-1 are required in the excretory cell for lumen extension. (**A**) Schematics of L4 larval stage worms depicting excretory cell-specific protein depletion using *excP::zif-1*. The H-shaped excretory canal is outlined and a hypothetical ubiquitous ZF1-tagged protein is depicted in green. The typical region of the canal examined by microscopy is enlarged to show cytoplasmic (yellow, *excP*::YFP) and lumenal membrane (cyan, IFB-1::CFP) markers used for analyzing excretory canal morphology. Anterior left, dorsal top. (**B and C**) L4 stage excretory canal in transgenic control (**B**) and *excP::zif-1* (**C**) animals expressing ZF1::GFP::CDC-42. Outline of excretory canal cytoplasm is indicated by dotted line. ZF1::GFP::CDC-42 is degraded in the excretory cell, but not surrounding cells (arrowhead), in *excP::zif-1* animals. (**D**) Endogenous expression of SEC-5::ZF1::YFP at the excretory canal lumenal membrane of L4 stage larva. (**E–J''**) Larval excretory canal phenotypes in control (**E–F''**), SEC-5$^{exc(-)}$ (**G–H''**), and RAL-1$^{exc(-)}$ (**I–J''**). Canal cytoplasm and lumenal membrane are marked by an extrachromosomal array expressing excretory cell-specific cytoplasmic and lumenal membrane markers (see panel A). Confocal images were acquired using ×20 (**E, G, I**) and ×63 objectives (**F–F''**, **H–H''**, **J–J''**). Excretory cell body indicated by asterisk. Posterior tip of excretory canal indicated by white arrow. Posterior excretory canal that has extended beyond the focal plane is indicated by dashed white arrow. Dashed box indicates approximate region represented in high magnification images. Outline of each animal is indicated by solid white line. Scale bars, 10 μm.

The online version of this article includes the following figure supplement(s) for figure 1:

**Figure supplement 1.** *t28h11.8p* is an excretory cell-specific promoter during embryonic and larval canal outgrowth.

connected by small bridges. In addition, we note that the size of cysts could be affected by swelling of the lumen as an indirect consequence of poor osmoregulation.

We measured canal length by examining where the posterior canal lumens ended relative to body length in L1 and L4 larvae, as these stages represent active outgrowth (L1) and maintenance (L4) of the canal lumen. Dividing the body into quartiles along its anterior-posterior axis, nearly all control larvae extended canals to the third quartile (51–75% of body length) at the L1 stage and the fourth quartile (76–100% of body length) by the L4 stage (*Figure 2*). However, in both SEC-5$^{exc(-)}$ and RAL-1$^{exc(-)}$ larvae, canal lumen length was significantly reduced at both L1 and L4 stages, with nearly all larvae containing canal lumens that extended to less than 50% body length (*Figure 2*). The

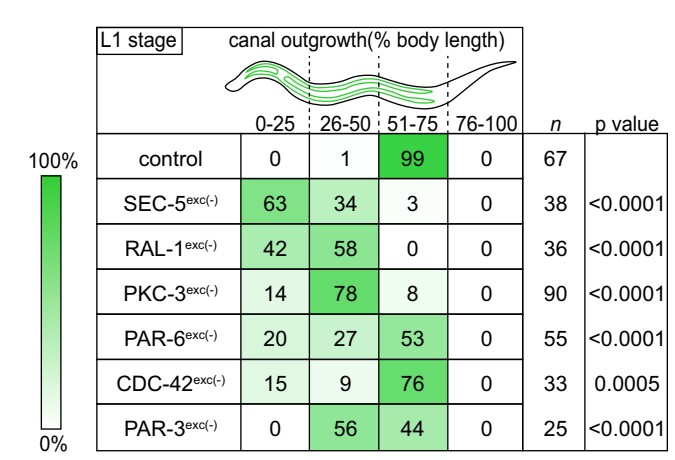

**Figure 2.** Canal outgrowth phenotypes upon exocyst or PAR protein depletion. Schematics of the excretory cell are shown at the L1 stage, when the canal is extending, and the L4 larval stage, when the canal is fully extended. Canal outgrowth defects upon depleting the indicated proteins in the excretory cell are depicted as the percentage of animals in each of four phenotypic categories (quartiles) that measure posterior canal extension relative to body length. The relative intensity of green shading reflects the percentage of larvae observed in each phenotypic category. p values were calculated using Fisher's exact test after pooling quartiles and comparing each genotype to the control group (L1 stage:<50% versus >50% canal outgrowth; L4 stage:<75% versus >75% canal outgrowth). p value significance was adjusted using Bonferroni correction to account for multiple comparisons to a common control, such that $p \leq 0.008$ is considered statistically significant.

The online version of this article includes the following source data and figure supplement(s) for figure 2:

**Source data 1.** Positions of posterior excretory canal arms in control, SEC-5[exc(-)], RAL-1[exc(-)], PKC-3[exc(-)], PAR-6[exc(-)], CDC-42[exc(-)], and PAR-3[exc(-)].

**Figure supplement 1.** The SEC-5[exc(-)] canal outgrowth phenotype is not enhanced by a *sec-5* null allele.

canal lumen length defect of SEC-5[exc(-)] larvae did not become more severe when we replaced one *sec-5(xn51: sec-5::zf1::yfp)* allele with the *sec-5(tm1443)* predicted null allele (*Frische et al., 2007*; *Figure 2—figure supplement 1*), suggesting that SEC-5[exc(-)] phenotypes result from nearly complete or complete loss of SEC-5 protein once the *excP::zif-1* transgene is expressed. Together, these data indicate that exocyst activity within the excretory cell is needed for proper organization and extension of its intracellular lumen.

## PAR proteins and CDC-42 are expressed in the excretory cell and have distinct localization patterns

We next addressed whether PAR proteins are required for extension of the excretory cell lumen using endogenously tagged alleles of *par-3, par-6,* and *pkc-3* expressing fusion proteins tagged with ZF1 and either YFP or GFP. *par-3::zf1::yfp* (this study), *par-6::zf1::yfp* (*Zilberman et al., 2017*) and *zf1::gfp::pkc-3* (*Montoyo-Rosario et al., 2020*) knock-in alleles were functional, as they did not cause the embryonic lethality (*Kemphues et al., 1988*; *Tabuse et al., 1998*; *Watts et al., 1996*) associated with *par-3, par-6,* or *pkc-3* inactivation (see Materials and methods) (*Montoyo-Rosario et al., 2020*; *Zilberman et al., 2017*). PAR-3::ZF1::YFP, PAR-6::ZF1::YFP, and ZF1::GFP:: PKC-3 proteins each concentrated at the excretory cell lumenal membrane within puncta (*Figure 3A–C*), similar to SEC-5::ZF1::YFP (*Figure 1D*).

The localization of CDC-42 within the excretory cell has only been described using high-copy transgenes and heterologous promoters (*Lant et al., 2015*; *Mattingly and Buechner, 2011*), and the high-copy transgene expressing ZF1::GFP::CDC-42 that we used to test the efficacy of *excP::zif-1* (*Figure 1B*; *Armenti et al., 2014b*). We examined CDC-42 subcellular localization in the excretory cell using a functional endogenously tagged *zf1::yfp::cdc-42* allele (*Zilberman et al., 2017*). ZF1:: YFP::CDC-42 protein was expressed in the excretory cell and showed a broader distribution than PAR-6::mKate (*Figure 3G–G''*). ZF1::YFP::CDC-42 extended well into the excretory cell cytoplasm compared to endogenously expressed PAR-6::mKate present within the same animal (*Figure 3D,G, H*), whereas endogenously tagged PAR-6::ZF1::YFP and PAR-3::mCherry showed a similar

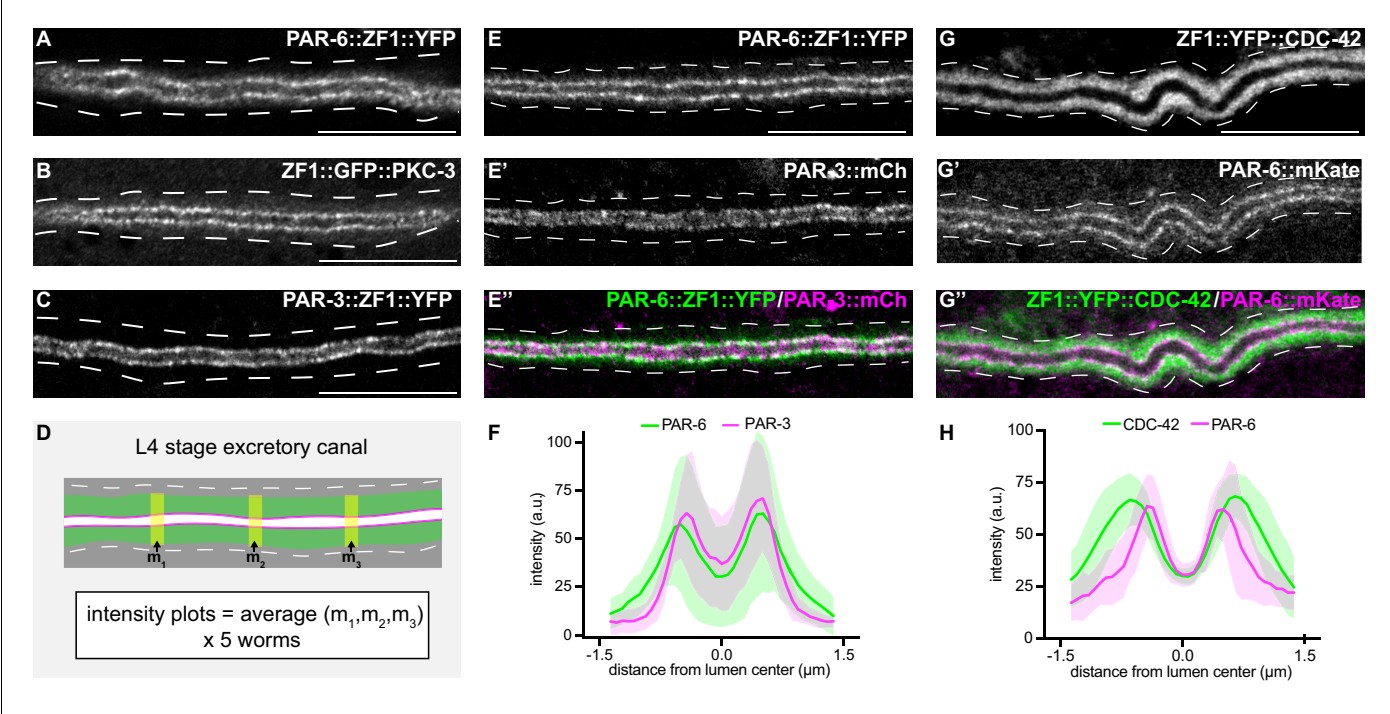

**Figure 3.** PAR-6, PKC-3, and PAR-3 are enriched at the lumenal membrane and CDC-42 extends into the canal cytoplasm. (A–C) Distribution of endogenously tagged PAR-6, PKC-3, and PAR-3 in the excretory cell canal. (D) Schematic of excretory cell line trace measurements displayed in **F** and **H**. Three line-trace measurements ($m_1$, $m_2$, $m_3$) were taken perpendicular to the excretory cell lumen in each animal. Measurements were averaged to generate a single line trace for each larva, and five larvae were measured from each genotype. (E–E′′) Distribution of PAR-6::ZF1::YFP and PAR-3:: mCherry in the larval excretory canal. (F) Line traces of PAR-6::ZF1::YFP (green) and PAR-3::mCherry (magenta). Solid line represents mean and shaded area is ± SD. Intensities were normalized to compare peak values of each channel. '0.0' on x-axis represents the center point of the canal lumen. n = 5 larvae. (G–G′′) Distribution of ZF1::YFP::CDC-42 and PAR-6::mKate in the larval excretory canal. (H) Line trace of ZF1::YFP::CDC-42 (green) and PAR-6:: mKate (magenta). Solid line represents mean and shaded area is ± SD. Intensities were normalized to compare peak values of each channel. '0.0' on x-axis represents the center point of the canal lumen. n = 5 larvae. Outline of excretory canal cytoplasm is indicated by dashed lines. Scale bars, 10 µm. The online version of this article includes the following source data for figure 3:

**Source data 1.** Fluorescent intensity values for line trace measurements of PAR-6::ZF1::YFP; PAR-3::mCherry and ZF1::YFP::CDC-42; PAR-6::mKate.

enrichment to the lumenal membrane (*Figure 3E–F*). While the peak localization intensities of ZF1:: YFP::CDC-42 and PAR-6::mKate in transects across the width of the excretory cell do not align, as they do with PAR-6::ZF1::YFP and PAR-3::mCherry, super-resolution imaging would be required to determine whether ZF1::YFP::CDC-42 is present at the lumenal domain. Therefore, consistent with previous findings made using immunostaining and transgenes (*Armenti et al., 2014a*), endogenously tagged PAR-3, PAR-6, and PKC-3 are each expressed within the excretory cell and are present at the lumenal membrane, and CDC-42 is expressed more broadly within the cytoplasm.

## PAR-6, PKC-3, and CDC-42 are required in the excretory cell for lumen extension

To determine if PAR proteins and CDC-42 are required within the excretory cell for lumen extension, we crossed *excP::zif-1* with each *par* or *cdc-42* knock-in allele and examined excretory canal morphology using cytoplasmic and lumenal membrane markers (see *Figure 1A*). PAR-6[exc(-)] and PKC-3[exc(-)] L4 stage larvae had severely truncated canals with dilated and cystic lumens (*Figure 4A–D′′*), similar to SEC-5[exc(-)] and RAL-1[exc(-)] larvae (see *Figure 1G–J′′*). CDC-42[exc(-)] larvae showed similar lumen extension defects (*Figure 4E–F′′*), but in addition some animals had a split-canal phenotype whereby two lumenized canals split from a single canal arm (n = 42/158 L4 larvae, *Figure 4—figure supplement 1*). Similar to SEC-5[exc(-)] and RAL-1[exc(-)] larvae, the length of the excretory canals was significantly shorter in PAR-6[exc(-)], PKC-3[exc(-)], and CDC-42[exc(-)] compared to controls at both the L1

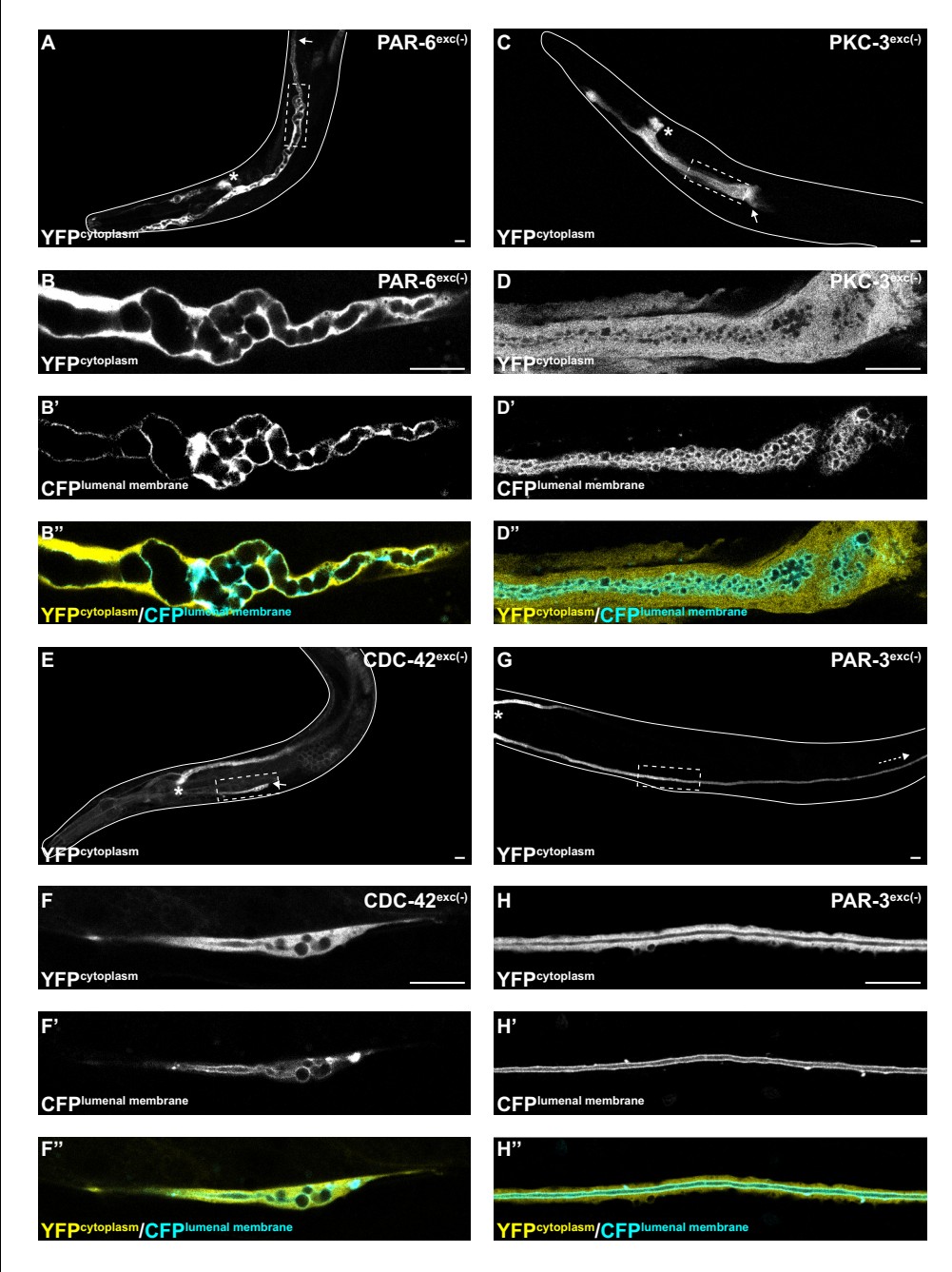

**Figure 4.** PAR-6, PKC-3, and CDC-42, but not PAR-3, are required for excretory cell lumen extension. Larval excretory canal phenotypes in PAR-6$^{exc(-)}$ (A–B''), PKC-3$^{exc(-)}$ (C–D''), CDC-42$^{exc(-)}$ (E–F'') and PAR-3$^{exc(-)}$ (G–H'') L4 stage worms expressing cytoplasmic and lumenal membrane markers. Confocal images were acquired using ×20 (A, C, E, G) and ×63 (B–B'', D–D'', F–F'', H–H'') objectives. Excretory cell body indicated by asterisk. Posterior tip of excretory canal indicated by white arrow. Posterior excretory canal that has extended beyond the focal plane is indicated by dashed white arrow. Dashed box indicates approximate region represented in high-magnification images. Outline of each animal is indicated by solid white line. Scale bars, 10 μm.

The online version of this article includes the following figure supplement(s) for figure 4:

**Figure supplement 1.** CDC-42 depletion causes a split lumen phenotype in larval excretory canals.

**Figure supplement 2.** Depletion of PAR-3 causes mild excretory cell lumen defects during early larval stages.

and L4 stages (*Figure 2*). Unexpectedly, PAR-3$^{exc(-)}$ larvae had a distinct and comparatively mild phenotype. At the L1 stage, canal lumens in PAR-3$^{exc(-)}$ larvae had an irregular diameter (*Figure 4—figure supplement 2*), and were significantly shorter than controls (*Figure 2*). However, by the L4 stage, the canals of PAR-3$^{exc(-)}$ larvae resembled those of controls (*Figure 4G–H''*) and were not significantly shorter (*Figure 2*). Although the phenotype of PAR-3$^{exc(-)}$ larvae appears distinct, more subtle differences in excretory canal length following the depletion of specific proteins might reflect variation in degradation rates or efficiency (*Nance and Frøkjær-Jensen, 2019*). All together, these findings suggest that PAR-6, PKC-3, and CDC-42 function within the excretory cell to promote extension of the lumen. PAR-3 is likely only important for lumen outgrowth during early stages, although we cannot exclude the possibility that an undescribed isoform of *par-3* with a different 3' end, and thus lacking the ZF1 tag, is expressed within the excretory cell and buffers mutant phenotypes. Our findings also show that, in addition to promoting lumen extension, CDC-42 functions to prevent canal arms from bifurcating.

## PAR-6, but not PAR-3, is required for exocyst lumenal membrane localization

The results above suggest that exocyst function or localization may require PAR-6, PKC-3, and CDC-42, but not PAR-3. To determine if PAR proteins regulate lumen extension by recruiting exocyst to the lumenal membrane, we acutely degraded PAR-6::ZF1::YFP and PAR-3::ZF1::YFP protein at the L4 larval stage, after canal growth was complete, by expressing ZIF-1 from a heat-shock promoter. This approach allowed us to analyze exocyst localization in anatomically normal canals, immediately after rapid PAR protein depletion (*Figure 5A*). Following a 30 minute heat shock to induce ZIF-1 expression at the L4 stage, PAR-6::ZF1::YFP degraded rapidly within 1 hour (*Figure 5B–C*, *Figure 5—figure supplement 1*). To monitor exocyst localization after PAR-6::ZF1::YFP depletion, we utilized a transgene expressing mCherry::SEC-10 (*Armenti et al., 2014a*), which like SEC-5::ZF1::YFP enriches at the lumenal membrane (*Figure 5B',D*). After PAR-6::ZF1::YFP degraded, mCherry::SEC-10 was no longer enriched at the lumenal membrane, but instead, appeared evenly distributed throughout the cytoplasm (*Figure 5C',E*). We quantified these changes in localization by comparing mCherry::SEC-10 intensity along the lumenal membrane to that within the adjacent cytoplasm by generating a lumen/cytoplasm intensity ratio (*Figure 5A*), which was significantly reduced in PAR-6-depleted larvae (*Figure 5F*). We performed analogous experiments to determine the role of PAR-3 in exocyst localization. In contrast to PAR-6::ZF1::YFP depletion, loss of PAR-3::ZF1::YFP did not decrease the enrichment of mCherry::SEC-10 at the lumenal membrane, despite a lack of visible PAR-3::ZF1::YFP protein following ZIF-1 induction (*Figure 5G–K*). We conclude that PAR-6 is required to enrich the exocyst complex at the lumenal membrane, whereas PAR-3 is likely dispensable for exocyst lumenal membrane enrichment.

## PAR-3 promotes PAR-6 lumenal membrane localization

In many polarized cell types, PAR-3 helps enrich PAR-6 at the membrane (*Nance and Zallen, 2011*; *St Johnston and Ahringer, 2010*). Therefore, the requirement for PAR-6, but not PAR-3, in mCherry::SEC-10 lumenal membrane enrichment was surprising. To investigate the epistatic relationship between PAR-3 and PAR-6 within the excretory cell, we first expressed ZIF-1 from a heat shock promoter and degraded PAR-3::ZF1::YFP after canal growth was complete (*Figure 6A–B*). Surprisingly, endogenously tagged PAR-6::mKate (*Dickinson et al., 2017*) was significantly less enriched at the lumenal membrane and increased within the cytoplasm after depletion of PAR-3::ZF1::YFP when compared to control larvae (*Figure 6A'–E*), although some puncta of PAR-6::mKate remained at the lumenal membrane (*Figure 6B'*, arrowheads). In reciprocal experiments, we degraded PAR-6::ZF1::YFP by expressing ZIF-1 from a heat shock promoter and examined endogenously tagged PAR-3::mCherry localization. PAR-3::mCherry remained enriched at the lumenal membrane in PAR-6-depleted L4 worms, and unexpectedly, its lumen/cytoplasm ratio was significantly increased (*Figure 6F–J*). We propose that PAR-3 is required to recruit most PAR-6 to the lumenal membrane, but that the PAR-6 puncta remaining after PAR-3::ZF1::YFP depletion are sufficient to recruit the exocyst to the lumenal membrane (see Discussion). In addition, these findings show that PAR-6 limits PAR-3 lumenal membrane enrichment.

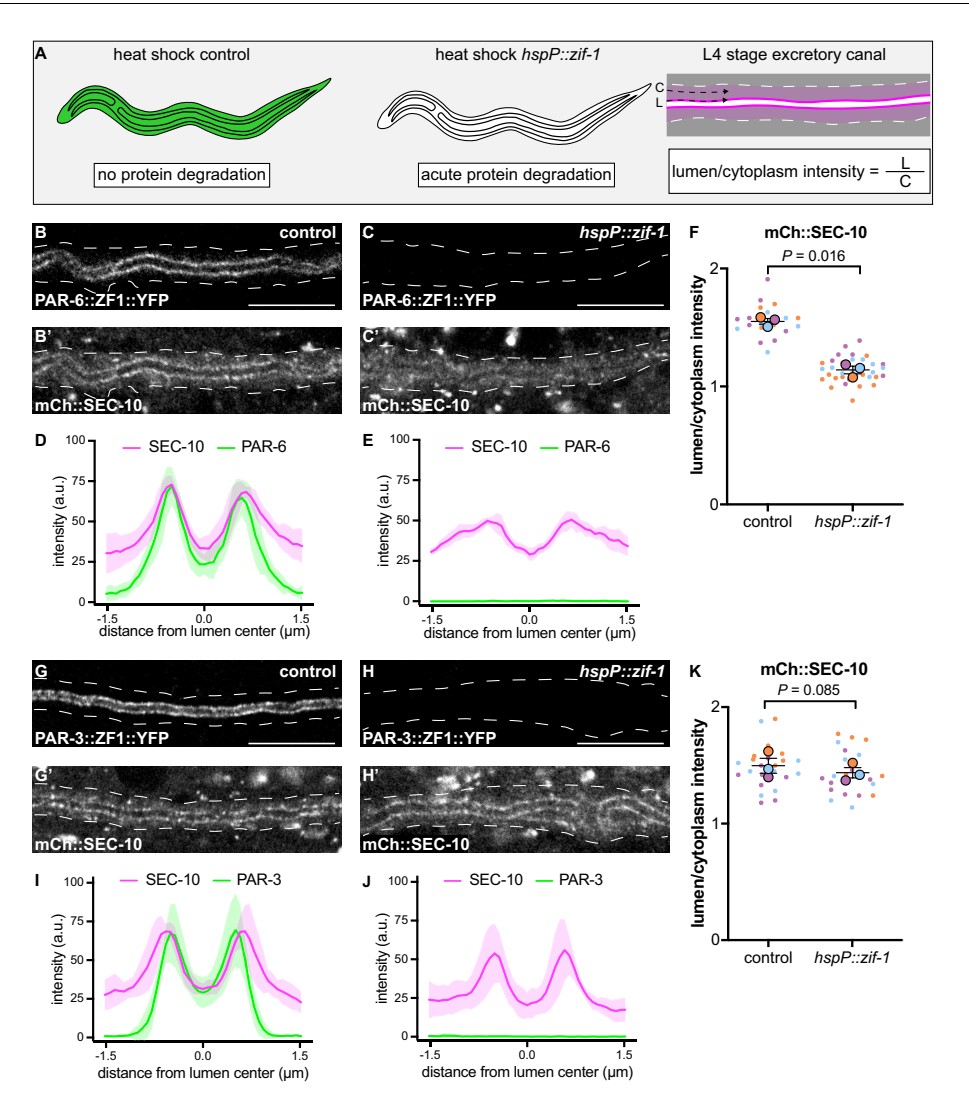

**Figure 5.** PAR-6, but not PAR-3, is required to enrich SEC-10 at the lumenal membrane. (**A**) Schematic of L4 larval stage worms depicting heat-shock inducible protein depletion. The excretory canal is outlined in black and a hypothetical ubiquitous ZF1-tagged protein is shown in green. Upon heat-shock, the ZF1-tagged protein is rapidly degraded in all somatic cells of animals expressing *hspP::zif-1*. To measure fluorescence intensity, average pixel intensity was calculated along a region of the excretory cell lumenal membrane ('L') and within the cytoplasm ('C'); dividing L/C yields the lumen/cytoplasm ratio shown in (**F** and **K**). Anterior left, dorsal top. (**B–C**) Distribution of PAR-6::ZF1::YFP in larval excretory canal in control (**B**) and *hspP::zif-1* (**C**). (**B'–C'**) Distribution of mCherry::SEC-10 in larval excretory canal of control (**B'**) and *hspP::zif-1* (**C'**) worms expressing PAR-6::ZF1::YFP. (**D–E**) Line trace of PAR-6::ZF1::YFP (green) and mCherry::SEC-10 (magenta). Intensities were normalized to compare peak values of each channel. '0.0' on x-axis represents the center point of the canal lumen. *n* = 5 larvae. (**F**) Quantification of lumenal membrane to cytoplasm intensity ratio of mCherry::SEC-10 in the excretory canal of control and *hspP::zif-1* larvae expressing PAR-6::ZF1::YFP. Individual data points (small dots) are color-coded (orange, purple, and light blue) from three independent replicates. Large dots represent the mean of each replicate, horizontal bar is the mean of means, and error bars are the SEM. p values were calculated using a ratio paired t-test of the means. *n* = 5, 8, 7 for control; *n* = 13, 11, 10 for *hspP::zif-1*. (**G–H**) Distribution of PAR-3::ZF1::YFP in larval excretory canal in control (**G**) and *hspP::zif-1* (**H**). (**G'–H'**) Distribution of mCherry::SEC-10 in the larval excretory canal of control (**G'**) and *hspP::zif-1* (**H'**) worms expressing PAR-3::ZF1::YFP. (**I–J**) Line trace of PAR-3::ZF1::YFP (green) and mCherry::SEC-10 (magenta). Intensities were normalized to compare peak values of each channel. '0.0' on x-axis represents the center point of the canal lumen. *n* = 5 larvae. (**K**) Quantification of lumenal membrane to cytoplasm intensity ratio of mCherry::SEC-10 expression in the excretory canal of control and *hspP::zif-1* larvae expressing PAR-3::ZF1::YFP. Data is shown as in panel F. p values were calculated using a ratio paired t-test of the means. *n* = 7, 9, 8 for

*Figure 5 continued*

control; *n* = 7, 8, 8 for *hspP::zif-1*. Outline of excretory canal cytoplasm is indicated by dashed line. Scale bars, 10 μm.

The online version of this article includes the following source data and figure supplement(s) for figure 5:

**Source data 1.** Fluorescent intensity values for line trace measurements of PAR-6::ZF1::YFP; mCherry::SEC-10 and PAR-3::ZF1::YFP; mCherry::SEC-10.

**Source data 2.** Fluorescent intensity values for lumenal membrane and cytoplasmic mCherry::SEC-10 measurements in PAR-6::ZF1::YFP and PAR-3::ZF1::YFP backgrounds.

**Figure supplement 1.** PAR-6::ZF1::YFP depletion by acute ZIF-1 expression.

## CDC-42 is required for PAR-6 lumenal membrane localization

We next asked what other factors act upstream to regulate the lumenal membrane enrichment of PAR-6 and PKC-3 within the excretory cell. One candidate is CDC-42, which binds to the PAR-6 CRIB domain and can recruit PAR-6 to the membrane in parallel to PAR-3 in the one-cell *C. elegans* embryo (*Aceto et al., 2006*; *Beers and Kemphues, 2006*; *Gotta et al., 2001*; *Joberty et al., 2000*; *Kay and Hunter, 2001*; *Rodriguez et al., 2017*; *Wang et al., 2017*). CDC-42$^{exc(-)}$ and PAR-6$^{exc(-)}$ larvae displayed a similar canal outgrowth phenotype (*Figure 2*), consistent with these two proteins acting in the same lumen extension pathway within the excretory cell. To determine if CDC-42 is required for PAR-6 enrichment at the lumenal membrane, we acutely degraded ZF1::YFP::CDC-42 by heat shock expression of ZIF-1 in L4 larvae. PAR-6::mKate lumenal membrane enrichment was significantly decreased after loss of CDC-42 (*Figure 7A–E*). Together, these results suggest that CDC-42 promotes lumen extension by helping to enrich PAR-6 at the lumenal membrane.

## EXC-5, a putative CDC-42 RhoGEF, is required for PKC-3 lumenal membrane localization

Given that only active GTP-bound CDC-42 interacts with PAR-6 (*Aceto et al., 2006*; *Gotta et al., 2001*), we hypothesized that CDC-42 at the lumenal membrane is activated by one or more RhoGEFs. EXC-5 is an orthologue of the faciogenital dysplasia-associated (FGD) family of RhoGEFs that can activate Cdc42 in biochemical and cell culture assays (*Hayakawa et al., 2008*; *Huber et al., 2008*; *Kurogane et al., 2012*; *Miyamoto et al., 2003*; *Steenblock et al., 2014*; *Umikawa et al., 1999*; *Zheng et al., 1996*), and EXC-5 has been proposed as an activator of CDC-42 in the excretory cell. *C. elegans* EXC-5::GFP over-expressed from a high-copy transgene is present within the excretory cell (*Mattingly and Buechner, 2011*; *Suzuki et al., 2001*), and *exc-5* mutants have shortened excretory cell canals. In addition, genetic epistasis experiments are consistent with *cdc-42* functioning downstream of *exc-5* (*Mattingly and Buechner, 2011*; *Shaye and Greenwald, 2016*). To determine whether EXC-5 is required for PAR-6 or PKC-3 protein localization, as is CDC-42, we created an endogenously tagged *exc-5* allele expressing EXC-5::ZF1::mScarlet. Like PAR-6 and PKC-3, EXC-5::ZF1::mScarlet was enriched at the lumenal membrane (*Figure 7F–F',H*). We used heat-shock inducible ZIF-1 to remove EXC-5::ZF1::mScarlet acutely and examined the effect on endogenously tagged GFP::PKC-3 (*Rodriguez et al., 2017*; *Wang et al., 2017*). Upon depletion of EXC-5::ZF1::mScarlet, GFP::PKC-3 enrichment at the lumenal membrane was significantly reduced compared to control larvae (*Figure 7G–J*). These results indicate that EXC-5 is required for PKC-3 recruitment to the excretory cell lumenal membrane, most likely through its activation of CDC-42.

## Discussion

### An intracellular lumenogenesis pathway bridging Rho GTPase, cell polarization, and vesicle-tethering proteins

During tubulogenesis within the *C. elegans* excretory cell, it has been proposed that the docking and subsequent fusion of large 'canalicular' vesicles at the lumenal membrane domain provides the membrane needed for tube extension (*Khan et al., 2013*; *Kolotuev et al., 2013*). We showed

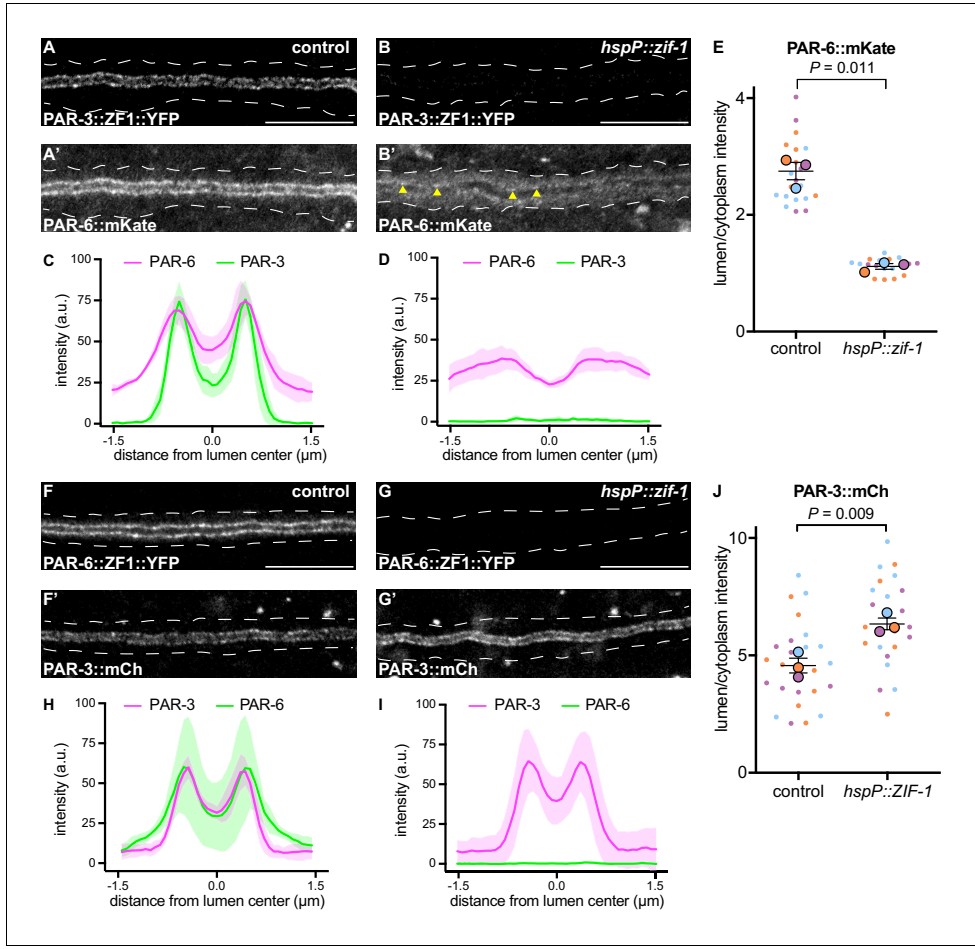

**Figure 6.** PAR-3 is required to enrich PAR-6 at the lumenal membrane. (**A–B**) Distribution of PAR-3::ZF1::YFP in larval excretory canal in control (**A**) and *hspP::zif-1* (**B**) worms. (**A'–B'**) Distribution of PAR-6::mKate in the larval excretory canal of control (**A'**) and *hspP::zif-1* (**B'**) worms expressing PAR-3::ZF1::YFP. Arrowheads show punctate PAR-6::mKate along lumenal membrane. (**C–D**) Line traces of PAR-3::ZF1::YFP (green) and PAR-6::mKate (magenta). Intensities were normalized to compare peak values of each channel. '0.0' on x-axis represents the center point of the canal lumen. *n* = 5 larvae. (**E**) Quantification of lumenal membrane to cytoplasm intensity ratio of PAR-6::mKate expression in the excretory canal of control and *hspP::zif-1* larvae expressing PAR-3::ZF1::YFP. Individual data points (small dots) are color-coded (orange, purple, and light blue) from three independent replicates. Large dots represent the mean of each replicate, horizontal bar is the mean of means, and error bars are the SEM. p values were calculated using a ratio paired t-test of the means. *n* = 6, 6, 8 for control; *n* = 4, 7, 8 for *hspP::zif-1*. (**F–G**) Distribution of PAR-6::ZF1::YFP in larval excretory canal in control (**F**) and *hspP::zif-1* (**G**) worms. (**F'–G'**) Distribution of PAR-3::mCherry in larval excretory canal of control (**F'**) and *hspP::zif-1* (**G'**) worms expressing PAR-6::ZF1::YFP. (**H–I**) Line traces of PAR-6::ZF1::YFP (green) and PAR-3::mCherry (magenta). Intensities were normalized to compare peak values of each channel. '0.0' on x-axis represents the center point of the canal lumen. *n* = 5 larvae. (**J**) Quantification of lumenal membrane to cytoplasm intensity ratio of PAR-3::mCherry expression in the excretory canal of control and *hspP::zif-1* larvae expressing PAR-6::ZF1::YFP. Data depicted as in panel E. p values were calculated using a ratio paired t-test of the means. *n* = 9, 8, 9 for control; *n* = 7, 8, 9 for *hspP::zif-1*. Outline of excretory canal cytoplasm is indicated by dotted line. Scale bars, 10 μm.

The online version of this article includes the following source data for figure 6:

**Source data 1.** Fluorescent intensity values for line trace measurements of PAR-3::ZF1::YFP; PAR-6::mKate and PAR-6::ZF1::YFP; PAR-3::mCherry.

**Source data 2.** Fluorescent intensity values for lumenal membrane and cytoplasmic measurements of PAR-6:: mKate in PAR-3::ZF1::YFP background and PAR-3::mCherry measurements in PAR-6::ZF1::YFP background.

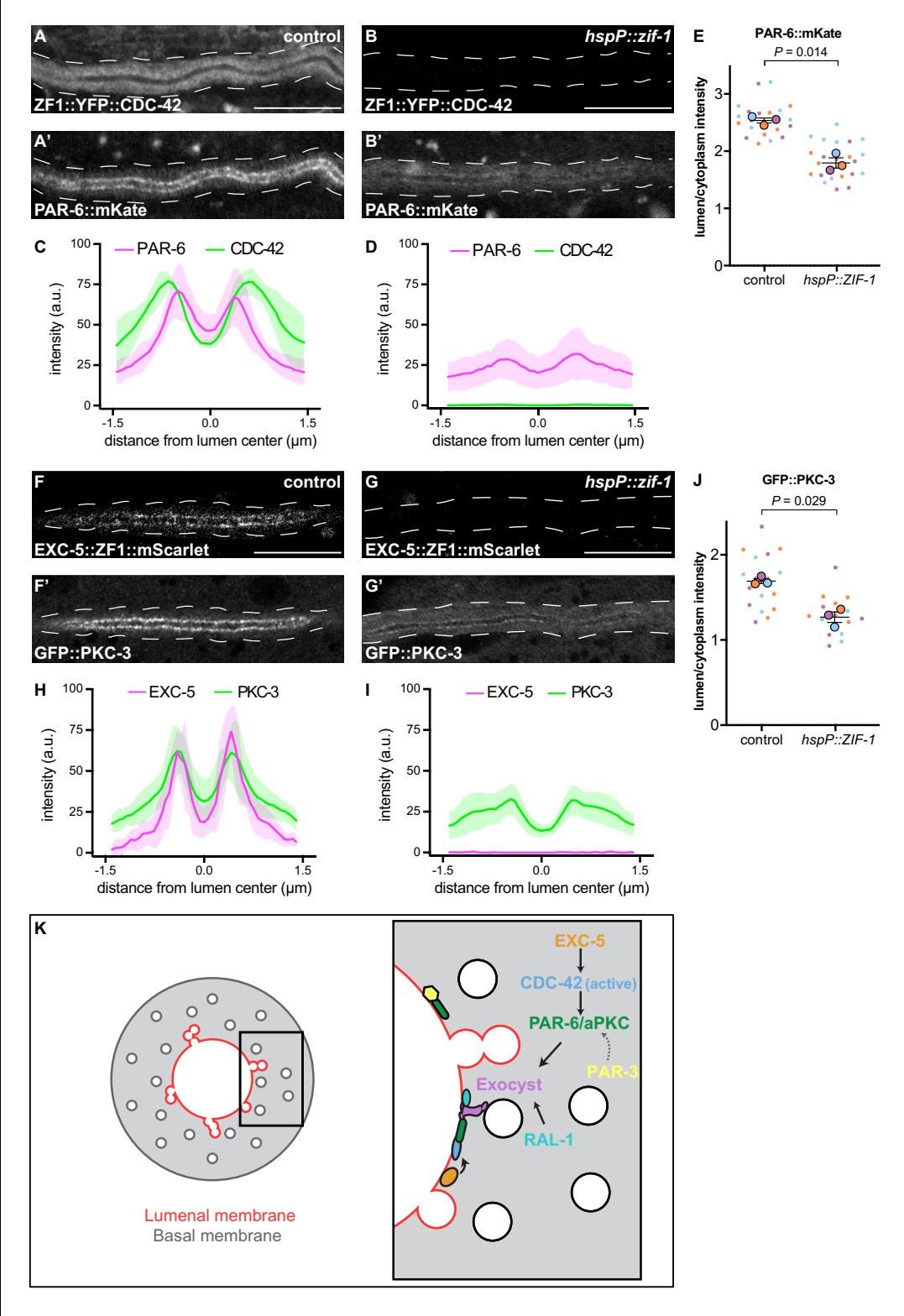

**Figure 7.** CDC-42 and EXC-5 are required to enrich PAR-6 and PKC-3 at the lumenal membrane. (**A–B**) Distribution of ZF1::YFP::CDC-42 in larval excretory canal in control (**A**) and *hspP::zif-1* (**B**) worms. (**A'–B'**) Distribution of PAR-6::mKate in the larval excretory canal of control (**A'**) and *hspP::zif-1* (**B'**) worms expressing ZF1:: YFP::CDC-42. (**C–D**) Line trace of ZF1::YFP::CDC-42 (green) and PAR-6::mKate (magenta). Intensities were normalized to compare peak values of each channel. '0.0' on x-axis represents the center point of the canal lumen.
*Figure 7 continued on next page*

*Figure 7 continued*

n = 5 larvae. (**E**) Quantification of lumenal membrane to cytoplasm intensity ratio of PAR-6::mKate expression in the excretory canal of control and *hspP::zif-1* larvae expressing ZF1::YFP::CDC-42. Individual data points (small dots) are color-coded (orange, purple, and light blue) from three independent replicates. Large dots represent the mean of each replicate, horizontal bar is the mean of means, and error bars are the SEM. p values were calculated using a ratio paired t-test of the means. n = 8, 7, 7 for control; n = 9, 7, 8 for *hspP::zif-1*. (**F–G**) Distribution of EXC-5::ZF1::mScarlet in the larval excretory canal in control (**F**) and *hspP::zif-1* (**G**) worms. (**F'–G'**) Distribution of GFP::PKC-3 in the larval excretory canal of control (**F'**) and *hspP::zif-1* (**G'**) worms expressing EXC-5::ZF1::mScarlet. (**H–I**) Line trace of GFP::PKC-3 (green) and EXC-5::ZF1::mScarlet (magenta). Intensities were normalized to compare peak values of each channel. '0.0' on x-axis represents the center point of the canal lumen. n = 5 larvae. (**J**) Quantification of lumenal membrane to cytoplasm intensity ratio of GFP::PKC-3 expression in the excretory canal of control and *hspP::zif-1* larvae expressing EXC-5::ZF1::mScarlet. Data are depicted as in panel E. p values were calculated using a ratio paired t-test of the means. n = 5, 6, 6 for control; n = 5, 5, 6 for *hspP::zif-1*. (**K**) Model of PAR and exocyst regulation of excretory cell lumen extension. Cross section of larval excretory canal (left) depicts large, canalicular vesicles fusing with the lumenal membrane (red) during lumen extension. Boxed region represents a portion of canal where lumen extension is occurring, magnified at right to show a proposed molecular pathway for lumenal vesicle tethering. Outline of excretory canal cytoplasm is indicated by dotted line. Scale bars, 10 μm.

The online version of this article includes the following source data for figure 7:

**Source data 1.** Fluorescent intensity values for line trace measurements of ZF1::YFP::CDC-42; PAR-6::mKate and EXC-5::ZF1::mScarlet; GFP::PKC-3.

**Source data 2.** Fluorescent intensity values for lumenal membrane and cytoplasmic measurements of PAR-6::mKate in ZF1::YFP::CDC-42 background and GFP::PKC-3 measurements in EXC-5::ZF1::mScarlet background.

---

previously that exocyst complex activity is required for canalicular vesicles to connect with the lumenal membrane domain and for normal lumen extension to occur (*Armenti et al., 2014a*). Here, based on cell-specific protein depletion experiments during lumen extension, and protein localization analysis following acute protein degradation in fully developed excretory cells, we propose a pathway for lumen extension (*Figure 7K*). Most upstream, RhoGEF EXC-5 at the lumenal membrane activates the Rho GTPase CDC-42. Although EXC-5 has been proposed previously as an activator of CDC-42 at the lumenal membrane (*Mattingly and Buechner, 2011*; *Shaye and Greenwald, 2016*), our findings show for the first time that its depletion causes a similar molecular defect as depletion of CDC-42 (loss of PKC-3 or PAR-6 from the lumenal membrane). Downstream of EXC-5, we propose that active CDC-42 recruits PAR-6 and PKC-3 through interactions with the PAR-6 CRIB domain. In turn, PAR-6 and PKC-3 function to recruit the exocyst. RAL-1 has previously been shown to promote exocyst membrane localization, including in the early *C. elegans* embryo (*Armenti et al., 2014a*). The strong phenotypes we observe in RAL-1[exc(-)] larvae suggest that RAL-1 has a similar function within the excretory cell.

Although PAR-6 and PKC-3 bind one another and are typically thought to function as an obligate pair, we note that our experiments do not directly address whether they function together in lumen extension. In addition, further experiments will be required to determine whether EXC-5 activates CDC-42 specifically at the lumenal membrane, as our model predicts, and to identify the biochemical links between EXC-5, CDC-42, PAR-6, PKC-3, and the exocyst complex.

Even though lumen extension is severely compromised in SEC-5[exc(-)], RAL-1[exc(-)], PAR-6[exc(-)], PKC-3[exc(-)], and CDC-42[exc(-)] larvae, the initial stages of lumenogenesis still occur. One possible explanation is that a distinct pathway directs the initial stages of lumen formation. Alternatively, since it is unclear whether the *excP::zif-1* transgene is active at the very early stages of lumenogenesis (see Results), it is possible that complete loss of the targeted proteins immediately after excretory cell birth would block lumen formation entirely. Finally, it is possible that degradation of the targeted ZF1-tagged proteins, while visibly below our level of detection by fluorescence, is not complete and phenotypes are hypomorphic. Resolving these possibilities will require the use of earlier-acting *zif-1* drivers and alternative genetic methods.

Although we found that in PAR-3-depleted larvae, most PAR-6 was lost from the excretory cell lumenal membrane – a phenotype that could be predicted based on previous studies of PAR-3 in other cell types – the relatively mild lumen extension phenotype of PAR-3$^{exc(-)}$ larvae (shortened canals in the L1 stage that recovered to normal length by the L4 stage) and lack of requirement for PAR-3 in mCherry::SEC-10 localization were somewhat surprising. Recently, using auxin-inducible protein degradation, it was shown that PAR-3 is not essential for *C. elegans* larval development, in contrast to PAR-6 and PKC-3 (*Castiglioni et al., 2020*). Although further experiments will be needed to determine if an alternative form of PAR-3 protein lacking the ZF1 degron is produced, we consider this unlikely, as no such isoforms have been described, and the loss of PAR-6 at the lumenal membrane suggests that PAR-3 depletion was effective. Instead, we favor the hypothesis that PAR-3 makes lumen extension more efficient by augmenting PAR-6 lumenal enrichment, and that partial PAR-6 recruitment by CDC-42 is sufficient for lumen extension. Studies in the zygote have shown that in addition to localizing PAR-6 and PKC-3 to the membrane, CDC-42 also promotes PKC-3 activity (*Rodriguez et al., 2017*), raising the possibility that it plays a more consequential role during lumen extension than PAR-3 by both localizing and activating the PAR-6/PKC-3 complex. Such a relationship between PAR-3 and CDC-42 in recruiting PAR-6 likely occurs in additional cell types, as PAR-3 depletion in the epidermis causes PAR-6 mislocalization but not the junction defects that occur following PAR-6 depletion in the same cells (*Achilleos et al., 2010*). While it is not yet clear why PAR-3 appears to be more important for lumen extension at earlier larval stages, this is when active lumen outgrowth occurs. A reasonable hypothesis is that partially compromised PAR-6 function (because of reduced enrichment at the lumenal membrane) may be more consequential at this stage of lumenogenesis.

*par-6*, *aPKC*, and the exocyst are also required for proper intracellular lumen growth in *Drosophila* tracheal cells (*Jones et al., 2014*), suggesting that this pathway may function as a general mechanism promoting intracellular tube extension. Notably, and consistent with our findings in the *C. elegans* excretory cell, mutations in *Drosophila baz (par-3)* do not prevent tracheal lumen extension, suggesting that in both cell types PAR-6 and PKC-3/aPKC perform the major role in exocyst regulation. PAR proteins and the exocyst are also required for organized lumen expansion in mammalian cell cysts grown in 3D culture (*Bryant et al., 2010*). Thus PAR-mediated exocyst recruitment to sites of lumen expansion, where additional membrane is needed, appears to be a feature common to both intracellular and multicellular tubes despite their dramatically different organization.

## Exocyst recruitment by PAR proteins

Together with previous studies, our findings suggest that PAR proteins and the exocyst may interface in multiple ways. In mammary epithelial cells, Par3 functions as an exocyst receptor, utilizing a lysine-rich domain to bind Exo70 and recruit the complex (*Ahmed and Macara, 2017*). However, in these cells, the exocyst also mediates membrane fusion at the basal membrane, where Par3 is not detected, suggesting that alternative exocyst receptors exist (*Ahmed et al., 2018*). Biochemical studies have also revealed interactions between the exocyst, PAR-6, and aPKC. For example, co-immunoprecipitation experiments in cultured rat kidney epithelial cells and in cortical neurons showed that aPKC immunoprecipitates with the exocyst proteins Sec8, Sec6, or Exo84 (*Lalli, 2009*; *Rosse et al., 2009*). Furthermore, Par6 can directly bind Exo84 in cultured mammalian neurons, and this interaction is promoted by the RAL-1 homologue RalA (*Das et al., 2014*). Finally, in rat kidney epithelial cells, aPKC helps recruit exocyst through the aPKC-interacting protein Kibra (*Rosse et al., 2009*). Together with these studies, our finding that PAR-6 but not PAR-3 is required to recruit SEC-10 to the lumenal membrane suggests that PAR-6 functions as an alternative means to recruit the exocyst complex to the membrane. Further studies will be needed to clarify whether it does so directly by functioning as an exocyst receptor, analogous to mammalian Par3 (*Ahmed and Macara, 2017*), or indirectly, for example through the kinase activity of aPKC. Because aPKC and Par6 localize interdependently in nearly all cell types examined, the fact that PKC-3$^{exc(-)}$ and PAR-6$^{exc(-)}$ larvae have similar lumen extension defects does not clarify how PKC-3 contributes to exocyst recruitment. Notably, *C. elegans* lacks a clear Kibra orthologue (*Yoshihama et al., 2012*), suggesting that if PKC-3 interfaces with the exocyst directly, it does so utilizing a distinct mechanism.

# Materials and methods

## Key resources table

| Reagent type (species) or resource | Designation | Source or reference | Identifiers | Additional information |
|---|---|---|---|---|
| Strain, strain background (*C. elegans*) | *xnIs23[cdc-42p::zf1::gfp::cdc-42 unc-119(+)]; unc-119(ed3)* | *Armenti et al., 2014b* | FT95 | Shown in *Figure 1B* |
| Strain, strain background (*C. elegans*) | *sec-5(tm1443)/mIn1[mIs14 dpy-10(e128)]* | *Frische et al., 2007* | FT1202 | Shown in *Figure 2—figure supplement 1*. See Genetic test of ZIF-1 degradation section in Materials and methods |
| Strain, strain background (*C. elegans*) | *sec-5(xn51[sec-5::zf1::yfp loxP unc-119(+) loxP]); unc-119(ed3)* | *Armenti et al., 2014b* | FT1523 | Shown in *Figure 1D* |
| Strain, strain background (*C. elegans*) | *xnIs23; xnEx437[t28h11.8p::mCherry, t28h11.8p::zif-1]; unc-119(ed3)* | This study | FT1692 | Shown in *Figure 1C*, *Figure 1—figure supplement 1*. See Transgene construction section in Materials and methods |
| Strain, strain background (*C. elegans*) | *par-3(xn59[par-3::zf1::yfp loxP unc-119(+) loxP]); unc-119(ed3)* | This study | FT1699 | Shown in *Figure 3C*. See CRISPR knock-ins section in Materials and methods |
| Strain, strain background (*C. elegans*) | *par-6(xn60[par-6::zf1::yfp loxP unc-119(+) loxP]); unc-119(ed3)* | *Zilberman et al., 2017* | FT1702 | Shown in *Figure 3A* |
| Strain, strain background (*C. elegans*) | *sec-5(xn51); xnIs547[t28h11.8p::zif-1]; par-3(it301[par-3::mCherry]); xnEx466[t28h11.8p::yfp::sl2::ifb-1::cfp, pRF4]* | This study | FT1834 | FT1523 crossed to FT1837. Shown in *Figure 1G-H''*, *Figure 2*, *Figure 2—figure supplement 1* |
| Strain, strain background (*C. elegans*) | *xnIs547; par-3(it301); xnEx466* | This study | FT1837 | Shown in *Figure 1E-F''*, *Figure 2* |
| Strain, strain background (*C. elegans*) | *par-6(xn60); xnIs547; xnSi31[sec-8p::sec-8::mCherry unc-119(+)]; xnEx473[t28h11.8p::yfp::sl2::ifb-1::cfp, pRF4]* | This study | FT1844 | Shown in *Figure 2*, *Figure 4A-B''* |
| Strain, strain background (*C. elegans*) | *par-3(xn59); xnIs547; xnSi31; xnEx475[t28h11.8p::yfp::sl2::ifb-1::cfp, pRF4]* | This study | FT1846 | Shown in *Figure 22*, *Figure 4G-H''*, *Figure 4—figure supplement 2* |
| Strain, strain background (*C. elegans*) | *cdc-42(xn65[zf1::yfp::cdc-42 loxP unc-119(+) loxP]); xnIs547; par-3(it301); xnEx477[t28h11.8p::yfp::sl2::ifb-1::cfp, pRF4]* | This study | FT1849 | Shown in *Figure 2*, *Figure 4E-F''*, *Figure 4—figure supplement 1* |

*Continued on next page*

Continued

| Reagent type (species) or resource | Designation | Source or reference | Identifiers | Additional information |
|---|---|---|---|---|
| Strain, strain background (*C. elegans*) | *ral-1(tm5205); xnIs472[ral-1p::zf1:: yfp::ral-1]; xnIs547;xnEx472[t28h11. 8p::yfp::sl2::ifb-1::cfp, pRF4]* | This study | FT1866 | Shown in *Figure 1I-J''*, *Figure 2* |
| Strain, strain background (*C. elegans*) | *pkc-3(xn84[zf1:: gfp::pkc-3]); xnIs547; xnEx466* | This study | FT1942 | *pkc-3(xn84)* crossed to FT1837 Shown in *Figure 2*, *Figure 4C-D''* |
| Strain, strain background (*C. elegans*) | *cdc-42(xn65); par-6(cp60[par-6:: mKate::3xMyc loxP unc-119(+) loxP]); xnEx481[hsp-16.41p::zif-1; t28h11.8p::yfp:: sl2::ifb-1::cfp, pRF4]* | This study | FT1945 | Shown in *Figure 3G-H* |
| Strain, strain background (*C. elegans*) | *par-3(xn59); par-6(cp60); xnEx491[t28h11. 8p::cfp, pRF4]* | This study | FT2015 | Shown in *Figure 6A-A',C,E* |
| Strain, strain background (*C. elegans*) | *par-6(xn60); par-3(it301); xnEx494[hsp-16.41p::zif-1; t28h11.8p::CFP, pRF4]* | This study | FT2020 | Shown in *Figure 6G-G',I,J*, *Figure 5—figure supplement 1* |
| Strain, strain background (*C. elegans*) | *par-6(xn60); par-3(it301); xnEx496[t28h11. 8p::CFP, pRF4]* | This study | FT2022 | Shown in *Figure 3E-F*, *Figure 6F-F',H,J* |
| Strain, strain background (*C. elegans*) | *par-3(xn59); par-6(cp60); xnEx501[hsp-16.41p::zif-1; t28h11.8p::CFP, pRF4]* | This study | FT2027 | Shown in *Figure 6B-B',D,E* |
| Strain, strain background (*C. elegans*) | *par-6(xn60); xnIs485[sec-10p:: mCherry::sec-10]; xnEx508[hsp-16.41p::zif-1; t28h11.8p::CFP, pRF4]* | This study | FT2061 | Shown in *Figure 5C-C',E,F* |
| Strain, strain background (*C. elegans*) | *par-6(xn60); xnIs485; xnEx511[t28h11. 8p::cfp, pRF4]* | This study | FT2065 | Shown in *Figure 5B-B',D,F* |
| Strain, strain background (*C. elegans*) | *par-3(xn59); xnIs485; xnEx514[ t28h11.8p::cfp, pRF4]* | This study | FT2069 | Shown in *Figure 5G-G',I,K* |
| Strain, strain background (*C. elegans*) | *exc-5(xn108[exc-5::zf1::mScarlet])* | This study | FT2074 | See CRISPR knock-ins section in Materials and methods |
| Strain, strain background (*C. elegans*) | *exc-5(xn108[exc-5::zf1::mScarlet]); pkc-3(it309[gfp::pkc-3])* | This study | FT2076 | FT2074 crossed to KK1228 |

*Continued*

| Reagent type (species) or resource | Designation | Source or reference | Identifiers | Additional information |
|---|---|---|---|---|
| Strain, strain background (*C. elegans*) | *exc-5(xn108); pkc-3(it309[gfp::pkc-3]); xnEx519[hsp-16.41p::zif-1; t28h11.8p::CFP, pRF4]* | This study | FT2089 | Shown in *Figure 7G-G',I,J* |
| Strain, strain background (*C. elegans*) | *exc-5(xn108); pkc-3(it309); xnEx523[ t28h11.8p::cfp, pRF4]* | This study | FT2093 | Shown in *Figure 7F-F',H,J* |
| Strain, strain background (*C. elegans*) | *par-3(xn59); xnIs485; xnEx528[hsp-16.41p::zif-1; t28h11.8p::CFP, pRF4]* | This study | FT2100 | Shown in *Figure 5H-H',J,KH* |
| Strain, strain background (*C. elegans*) | *cdc-42(xn65); par-6(cp60); xnEx551[ hsp-16.41p::zif-1; t28h11.8p::CFP, pRF4]* | This study | FT2289 | Shown in *Figure 7A-E* |
| Strain, strain background (*C. elegans*) | *par-3(it301)* | Gift from K. Kemphues (Cornell University, Ithaca, NY) | KK1218 | |
| Strain, strain background (*C. elegans*) | *pkc-3(it309)* | Gift from K. Kemphues (Cornell University, Ithaca, NY) | KK1228 | |
| Strain, strain background (*C. elegans*) | *par-6(cp60); par-3(cp54 [mNeonGreen:: 3xFlag::par-3])* | *Dickinson et al., 2017* | LP282 | |
| Recombinant DNA reagent | *Peft-3::Cas9 + ttTi5605 sgRNA* | *Dickinson et al., 2013* | pDD122 | Cas9 + sgRNA plasmid that is targeted to a genomic site near the ttTi5605 Mos1 insertion allele. Addgene plasmid #47550 |
| Recombinant DNA reagent | *t28h11.8p::mCherry* | This study | pJA022 | See transgene construction section in Materials and methods |
| Recombinant DNA reagent | *t28h11.8p::zif-1* | This study | pJA027 | See transgene construction section in Materials and methods |
| Recombinant DNA reagent | *Peft-3::Cas9 + par-3 sgRNA* 1 | sgRNA target sequence: GTACTGGGGAAAA CGATGAGG | pJA029 | Cas9 + sgRNA targeting genomic site at *par-3* locus. Derived from pDD122. |
| Recombinant DNA reagent | *Peft-3::Cas9 + par-3 sgRNA* 2 | sgRNA target sequence: GAAGCCTACGA GACACGTGG | pJA030 | Cas9 + sgRNA targeting genomic site at *par-3* locus. Derived from pDD122. |
| Recombinant DNA reagent | *Peft-3::Cas9 + par-6 sgRNA* 1 | sgRNA target sequence: GCACCGCAGC CGCTACAGG | pJA031 | Cas9 + sgRNA targeting genomic site at *par-6* locus. Derived from pDD122. *Zilberman et al., 2017* |
| Recombinant DNA reagent | *Peft-3::Cas9 + par-6 sgRNA* 2 | sgRNA target sequence: GTCCACCTGTAG CGGCTGCGG | pJA032 | Cas9 + sgRNA targeting genomic site at *par-6* locus. Derived from pDD122. *Zilberman et al., 2017* |

*Continued on next page*

Continued

| Reagent type (species) or resource | Designation | Source or reference | Identifiers | Additional information |
|---|---|---|---|---|
| Recombinant DNA reagent | *par-3::zf1::yfp + unc-119* | This study | pJA033 | Homologous repair plasmid for *par-3* with ten silent point mutations adjacent to sgRNA cut sites |
| Recombinant DNA reagent | *par-6::zf1::yfp + unc-119* | *Zilberman et al., 2017* | pJA034 | Homologous repair plasmid for *par-6* with six silent point mutations adjacent to sgRNA cut sites |
| Recombinant DNA reagent | *zf1::yfp::cdc-42 + unc-119* | *Zilberman et al., 2017* | pJA036 | Homologous repair plasmid for *cdc-42* with five silent point mutations adjacent to sgRNA cut sites |
| Recombinant DNA reagent | *Peft-3::Cas9 + cdc-42 sgRNA* | sgRNA target sequence: GTCACAGT AATGATCGG | pJA037 | Cas9 + sgRNA targeting genomic site at *cdc-42* locus. Derived from pDD122. *Zilberman et al., 2017* |
| Recombinant DNA reagent | *t28h11.8p:: ifb-1::cfp* | This study | pJA042 | See transgene construction section in Materials and methods |
| Recombinant DNA reagent | *t28h11.8p::yfp:: sl2::ifb-1::cfp* | This study | pJA043 | See transgene construction section in Materials and methods |
| Recombinant DNA reagent | *hsp-16.41p::zif-1* | This study | pJA045 | See transgene construction section in Materials and methods |
| Recombinant DNA reagent | *t28h11.8p::cfp* | This study | pJA050 | See transgene construction section in Materials and methods |
| Recombinant DNA reagent | *zf1::yfp + unc-119* | *Armenti et al., 2014b* | pJN601 | Plasmid backbone used to generate pJA033. Addgene plasmid #59790. |
| Recombinant DNA reagent | *pgp-12p::mCherry* | *Armenti et al., 2014b* | pSA086 | Plasmid backbone used to generate pJA022 |
| Recombinant DNA reagent | *hsp-16.41p::zif-1:: sl2::mCherry* | *Armenti et al., 2014b* | pSA120 | Plasmid backbone used to generate pJA045. Addgene plasmid #59789 |
| Recombinant DNA reagent | *Peft-3::Cas9 + sec-5 sgRNA* | sgRNA target sequence: gattatcg gctgtgttgta | pSA121 | Cas9 + sgRNA targeting genomic site at *sec-5* locus. Derived from pDD122. *Armenti et al., 2014b* |
| Recombinant DNA reagent | *sec-5::zf1::yfp + unc-119* | *Armenti et al., 2014b* | pSA122 | Homologous repair plasmid for *sec-5* with a silent point mutation in the sgRNA cut site |
| Sequence-based reagent | *exc-5(xn108) crRNA* | gaatcaTCATT CAGATTGCT | | crRNA (IDT) target site used to target the *exc-5* locus |

*Continued on next page*

Continued

| Reagent type (species) or resource | Designation | Source or reference | Identifiers | Additional information |
|---|---|---|---|---|
| Sequence-based reagent | exc-5(xn108)_F | CGAATGTACACAATGACCGCTGAAGACGAACAAACCCAAATGAAATGGTTGGCGATTTTGGATTTAGCCGCAAACGCA-CATCTGAAGAATCAACG-GAATTCTGGATCCGAACA-GAGCGAACCGACAGAATACAAAACGCGAC | | Forward primer for *zf1::mScarlet* dsDNA repair template with 120 bp homology arms. Includes five silent point mutations adjacent to predicted crRNA cut sites |
| Sequence-based reagent | exc-5(xn108)_R | gaaaatttggatacagtttcaacgaacgaataataagaattgagagaaaaacaagaatagaacactgaaataactaagaaaataaacatatgtcttggctgggtgccaaaaaagaatcaTCACTTGTAGAGCTCGTCCATTCCTC | | Reverse primer for *zf1::mScarlet* dsDNA repair template with 120 bp homology arms |
| Sequence-based reagent | t28h11.8p_F | atgtgggcgtgaacaaaaa | | Forward primer to amplify *t28h11.8p* from genomic DNA |
| Sequence-based reagent | t28h11.8p_R | tccagttgaaattgaac | | Reverse primer to amplify *t28h11.8p* from genomic DNA |
| Sequence-based reagent | par-3(xn59) 5′ homology arm_F | ACTTCCGGATATGAGTCGTACGCCGACTCTGAGCTC | | Forward primer to amplify *par-3* 5′ homology arm for Gibson cloning to generate pJA033 |
| Sequence-based reagent | par-3(xn59) 5′ homology arm_R | AGAGATCAGGGACCGCCGCACCGATTCCCT-CAGTAC | | Reverse primer to amplify *par-3* 5′ homology arm for Gibson cloning to generate pJA033. Includes five silent point mutations adjacent to predicted crRNA (pJA029) cut sites shown as underlined base pairs |
| Sequence-based reagent | par-3(xn59) 5′ homology arm | AACAAACTTCGGGGGAGAAGCCTATGAAACTCGAGGCGGAG-GAGCCGGC | | Forward + Reverse primer to generate five silent point mutations adjacent to predicted crRNA (pJA030) cut sites shown as underlined base pairs |
| Sequence-based reagent | par-3(xn59) 3′ homology arm_F | gtcagtttttctcaaagttatattacgcagcc | | Forward primer to amplify *par-3* 3′ homology arm for Gibson cloning to generate pJA033 |
| Sequence-based reagent | par-3(xn59) 3′ homology arm_R | gttgatagtattgtggaacgagacaatcc | | Reverse primer to amplify *par-3* 3′ homology arm for Gibson cloning to generate pJA033 |

*Continued on next page*

*Continued*

| Reagent type (species) or resource | Designation | Source or reference | Identifiers | Additional information |
|---|---|---|---|---|
| Software, algorithm | Fiji | GitHub | RRID:SCR_002285 | https://fiji.sc/ |
| Software, algorithm | GraphPad Prism 8 | GraphPad | RRID:SCR_002798 | https://www.graphpad.com/scientific-software/prism/ |
| Software, algorithm | Adobe Illustrator CC | Adobe Systems Inc | RRID:SCR_010279 | |

## *C. elegans* strains

Strains used in this study are listed in the Key Resources Table. All strains were cultured on Nematode Growth Medium (NGM) plates seeded with *Escherichia coli* OP50 bacteria and maintained at 20°C unless specified otherwise (*Brenner, 1974*).

## Transgene construction

All transgenes were constructed using Gibson assembly (*Gibson et al., 2009*) as follows:

pJA022 (*t28h11.8p::mCherry*) was assembled using vector pSA086 (*pgp-12p::mCherry, Armenti et al., 2014b*), and the *t28h11.8p* promoter was amplified from genomic DNA. 785 bp of sequence upstream of the start codon of *t28h11.8* gene was used to generate the *t28h11.8p* promoter.

pJA027 (*t28h11.8p::zif-1*) was assembled using vector pSA097 (*pgp-12p::zif-1*) containing *zif-1* coding sequence, and the *t28h11.8p* promoter sequence was added by Gibson assembly.

*t28h11.8p::yfp* and *ifb-1::cfp* were co-expressed in the same operon by inserting SL2 *trans*-splice acceptor sequences (244 bp intergenic sequence between *gpd-2* stop codon and *gpd-3* start site) between the *yfp* stop codon and the *ifb-1* start codon (*Tursun et al., 2009*). pJA043 (*t28h11.8p:: yfp::sl2::ifb-1::cfp*) was assembled using vector pJA042 (*t28h11.8p::ifb-1::cfp*) which contains *ifb-1* coding sequence; *yfp* and *sl2* were inserted between the promoter and *ifb-1*; *sl2* was amplified from pJN645. *yfp* (pPD136.64) and *cfp* (pPD136.61) have synthetic introns (Fire lab vector kit).

pJA045 (*hsp-16.41p::zif-1*) was assembled using vector pSA120 which contains *hsp-16.41* promoter sequence (*Armenti et al., 2014b*; *Hao et al., 2006*), and *zif-1* coding sequence was added by Gibson assembly.

pJA050 (*t28h11.8p::cfp*) was assembled using vector pJA027 (*t28h11.8p::zif-1*), and *cfp* was added by Gibson assembly.

## CRISPR knock-ins

Plasmids for CRISPR/Cas9 genomic editing to make *par-3(xn59[par-3::zf1::yfp loxP unc-119(+) loxP])* were constructed as described previously (*Dickinson et al., 2013*). The guide RNA sequence from plasmid pDD122 was replaced with the sequences (5'-GTACTGGGGAAAACGATGAGG-3') and (5'-GAAGCCTACGAGACACGTGG-3') to create two single guide RNAs (sgRNAs) that cleave near the *par-3* C-terminus (plasmids pJA029 and pJA030). A homologous repair plasmid for *par-3* (pJA033) was constructed using Gibson assembly. The following DNA segments were assembled in order: 1179 bp upstream of *par-3* stop codon (including ten silent point mutations adjacent to the predicted sgRNA cut sites) as the left homology arm; *zf1::yfp* with *unc-119*; and the 3' terminal 932 bp of *par-3* genomic sequence as the right homology arm. *zf1::yfp* with *unc-119* flanked by LoxP sites was amplified from plasmid pJN601, which contains LoxP-flanked *unc-119* inserted in reverse orientation into a synthetic intron within *yfp* (*Armenti et al., 2014b*). The vector backbone was PCR-amplified from pJN601 using Gibson assembly primers that overlapped with homology arms for *par-3*.

*par-3(xn59: par-3-zf1-yfp + unc-119)* was generated by microinjecting the sgRNA plasmids pJA029 and pJA030 (which also contains *Cas9*), the homologous repair template pJA033, and plasmid co-injection markers pGH8 (*rab-3P::mCherry::unc-54utr*; plasmid 19359; Addgene), pCFJ104

(*myo-3P::mCherry::unc-54utr*; plasmid 19328; Addgene), pCFJ90 (*myo-2P::mCherry::unc-54utr*; plasmid 19327; Addgene), and pMA122 (*peel-1* negative selection; plasmid 34873; Addgene) into *unc-119(ed3)* mutant worms (*Dickinson et al., 2013*; *Frøkjær-Jensen et al., 2012*). Plates containing non-Unc F2 transformants were heat-shocked at 34°C for 4 hr to activate PEEL-1 toxin in array-bearing animals, and successfully edited non-Unc animals were confirmed by the absence of mCherry expression in the F2 generation and YFP expression in their progeny.

*exc-5(xn108[exc-5::zf1::mScarlet])* was generated by injecting a crRNA (IDT) with target homology sequence (5'-GAATCATCATTCAGATTGCT-3'). *zf1::mScarlet* dsDNA repair template with ~120 bp homology arms was prepared using primers (5'-CGAATGTACACAATGACCGCTGAAGACGAA-CAAACCCAAATGAAATGGTTGGCGATTTTGGATTTAGCCGCAAACGCACATCTGAAGAA TCAACGGAATTCTGGATCCGAACAGAGCGAACCGACAGAATACAAAACGCGAC-3'),which included five silent point mutations adjacent to the predicted crRNA cut sites, and (5'-gaaaatttggata-cagtttcaacgaacgaataataagaattgagagaaaaacaagaatagaacactgaaataactaagaaaataaaca-tatgtcttggctgggtgccaaaaaagaatcaTCACTTGTAGAGCTCGTCCATTCCTC-3'), with plasmid pJA047 as a template. F1 worms with the co-CRISPR *dpy-10(cn64)* mutation (*Paix et al., 2016*) were screened by fluorescence and verified by PCR and sequencing.

Knock-in alleles were functional and viable, with only a minor level of lethality (*par-3(xn59)*, 97% [353/363] viable; *exc-5(xn108)*, 99% [400/405] viable).

## Transgene integration

pJA027 (*t28h11.8p::zif-1*), which contains an *unc-119(+)* transformation marker, was injected into *unc-119(ed3)* worms to obtain a stably inherited, high-copy extrachromosomal array. The array was integrated using Trioxsalen (Sigma) and UV irradiation. A mixed population of washed transgenic worms was incubated in 600 ml of 33.3 ng/ml Trioxsalen in DMSO in the dark for 15 min. Worms were dripped onto an unseeded NGM agar plate and, after the solution soaked in, the agar plate was irradiated with 360 μJ of UV light in a Stratalinker. NA22 bacterial food was dripped onto the worms and, after 5 hr in darkness, 20 L4 stage transgenic worms were picked to each of 20 peptone plates (10 cm) seeded with NA22 bacteria. F1 adults were bleached to collect eggs, which were plated 200 per plate onto 70 NGM plates (6 cm). Nine hundred eighty-four transgenic F2s were picked into individual wells of 24-well plates, and those with an F3 brood containing only non-Unc progeny were saved. Transgenic insertion *xnIs547* was isolated and outcrossed three times to *unc-119(ed3)*.

## Imaging

For all live-imaging experiments, larvae were mounted onto 5% agarose pads in a 2 mM Levamisole solution in M9 buffer to induce paralysis. Fluorescent images were acquired using an SP8 confocal microscope (Leica), 63 × 1.4 NA oil-immersion objective, 458, 488-, 514-, 561 nm lasers, and 1-5x zoom. For intensity measurements, larvae were imaged using HyD detectors and the photon-counting mode. Images were analyzed and processed in ImageJ (NIH) with no γ adjustments and level adjustments across pixels. For quantifications, the same laser power and exposure times were used within experiments and control and mutant images were processed similarly. After processing in ImageJ, images were rotated and cropped using Illustrator (CC2020, Adobe).

Fluorescence images for *Figure 1D*, *Figure 1—figure supplement 1*, and *Figure 4—figure supplement 1* were acquired on an Axio Imager.A2 microscope (Zeiss) with 63 × 1.4 NA or 40 × 1.3 NA objective and a CCD camera (model C10600-10B-H, S. 160522; Hamamatsu). Images were processed using the unsharpen mask method in ImageJ.

## Heat-shock expression of ZIF-1

Plates containing late L4/young adult animals were placed in a water bath at 34°C for 30 min and then transferred to 15°C to recover. In each experiment, control and experimental animals were imaged 2–4 hr following heat shock.

## Excretory canal outgrowth measurements

SEC-5exc(-), RAL-1exc(-), PKC-3exc(-), PAR-6exc(-), CDC-42exc(-), and PAR-3exc(-) strains were all homozygous viable when grown on NGM plates. Excretory canal length was scored visually using a canal-

specific cytoplasmic marker (*t28h11.8p::yfp*) at L1 and L4 larval stages. Both posterior canal arms were scored in each animal. In cases where the canal arms differed in length, an approximate average of the two lengths was recorded for that animal.

## Genetic test of ZIF-1 degradation

To generate SEC-5$^{exc(-)}$/*sec-5(tm1443)*: *sec-5(tm1443)/mln1* males were mated with *sec-5(xn51); xnls547[t28h11.8p::zif-1]* hermaphrodites that contain the *xnEx466* extrachromosomal array marking the canal lumen and cytoplasm. Canal length in *Figure 2—figure supplement 1* was scored in F1 generation male cross progeny that did not carry the *mln1* balancer [genotype was *sec-5(xn51)/sec-5(tm1443); xnls547[t28h11.8p::zif-1]/+*]. Controls were generated by mating *sec-5(xn51)* males with *sec-5(xn51); xnls547[t28h11.8p::zif-1]* hermaphrodites that carried the *xnEx466* extrachromosomal array. Canal length of controls was scored in F1 generation male cross progeny [genotype was *sec-5 (xn51); xnls547[t28h11.8p::zif-1]/+*].

## Image analysis

All measurements were performed using ImageJ and raw SP8 confocal image files. For lumen/cytoplasm intensity measurements, a line four pixels in width was drawn along the lumenal membrane and a second line was drawn along an adjacent region within the canal cytoplasm, as shown in *Figure 5A*. Mean pixel intensity values along each line were calculated using the ImageJ measuring tool. Both faces of the lumenal membrane were measured in each image and two images were acquired of different regions of the posterior canal arms within each animal. Four such measurements were taken for each animal and an average 'lumen/cytoplasm intensity ratio' was calculated, which is represented by small colored dots in plots in *Figures 5F, K*, *6E, J*, *7E and J*.

For intensity profiles of the excretory canal, a line 30 pixels in width was drawn across a 3 µm region of the excretory canal cytoplasm, as shown in *Figure 3D*. Three measurements were acquired for each animal and averaged to generate a single intensity profile per animal. Measurements from five animals are shown in each graph. Values were copied into GraphPad Prism 8 to generate an XY line plot displaying the average and standard deviation.

To measure excretory canal fluorescence intensity after ZIF-1 degradation, the polygon tool in ImageJ was used to draw a region of interest (ROI) around the canal cytoplasm using the CFP$^{cytoplasm}$ marker. Mean pixel intensity values within each polygon were calculated using the ImageJ measuring tool. To measure degradation, fluorescent intensity of PAR-6::ZF1::YFP was calculated in control and *hspP::zif-1* animals 2 hr after a 30 min heat shock at 34˚C. Two images were acquired of different regions of the posterior canal arms of each animal and averaged. Background YFP autofluorescence was calculated in wild type larvae carrying the *pgp-12p::mCherry* transgene to mark canal cytoplasm. Average background autofluorescence was subtracted from control and *hspP::zif-1* animals prior to calculating percent of YFP depletion. Error bars represent standard deviation, and were calculated from the change in mean fluorescence intensity between control and experimental animals.

For plotting image quantification and statistical analysis, mean values for each animal and each biological replicate were copied to GraphPad Prism 8. SuperPlots were generated in GraphPad Prism 8 as previously described (*Lord et al., 2020*), with dots of the same color representing individual data points from the same experiment.

## Statistics

Statistical analysis was performed in GraphPad Prism 8. Statistical tests, number of embryos, and number of experiments are indicated in the figure legends. No statistical tests were used to predetermine sample size. Animals were selected for measurements based on developmental stage, orientation on the slides, and health. No animals were excluded from analyses post-hoc. Investigators were not blinded to allocation during experiments and outcome assessment.

In *Figure 2*, data from quartiles was pooled into two categories and Fisher's exact test was then performed (see Figure Legend). Some categories (i.e. quartiles) contained small numbers (<10 larvae) which can cause the p value to be inaccurate for a test of independence and therefore pooling categories is appropriate in this instance (*McDonald, 2014*). Where multiple comparisons were made to a common control, p values were corrected using the Bonferroni method.

## Acknowledgements

We thank Ken Kemphues, Dan Dickinson, and Bob Goldstein for generous gifts of worm strains, Steve Armenti for plasmids used in transgene construction, and members of the Nance laboratory and Jane Hubbard for comments. Some strains were provided by the CGC, which is funded by NIH Office of Research Infrastructure Programs (P40 OD010440). This work was supported by fellowships from the American Cancer Society and the National Institutes of Health to JA (PF-16-175-01-DDC and F32HL136038) and research grants from the National Institutes of Health to JN (R01GM098492 and R35GM118081).

## Additional information

### Funding

| Funder | Grant reference number | Author |
| --- | --- | --- |
| American Cancer Society | PF-16-175-01-DDC | Joshua Abrams |
| National Institutes of Health | F32HL136038 | Joshua Abrams |
| National Institutes of Health | R01GM098492 | Jeremy Nance |
| National Institutes of Health | R35GM118081 | Jeremy Nance |

The funders had no role in study design, data collection and interpretation, or the decision to submit the work for publication.

### Author contributions

Joshua Abrams, Conceptualization, Formal analysis, Funding acquisition, Investigation, Visualization, Methodology, Writing - original draft, Writing - review and editing; Jeremy Nance, Conceptualization, Supervision, Funding acquisition, Methodology, Writing - review and editing

### Author ORCIDs

Joshua Abrams https://orcid.org/0000-0003-1834-6782
Jeremy Nance https://orcid.org/0000-0003-4212-7731

### Decision letter and Author response

Decision letter https://doi.org/10.7554/eLife.65169.sa1
Author response https://doi.org/10.7554/eLife.65169.sa2

## Additional files

### Supplementary files

• Transparent reporting form

### Data availability

All data generated or analyzed during this study are included in the manuscript and supporting files. Source data files have been provided for Figures 2, 3, 5, 6, and 7.

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
