## [Decision Letter]

[Editors' note: this paper was reviewed by Review Commons.]

**Acceptance summary:**

This paper reports the interesting discovery of a new molecular pathway of intracellular tubule extension. The recruitment of exocyst vesicles by apical proteins provides a mechanistic link between apical basal polarity and the subcellular insertion of membrane for tubule morphogenesis.

---

## [Author Response]

Please note the responses below address revisions after feedback from the Editors and *eLife*’s updated policy on revisions which asks ‘that the manuscript be revised to either limit claims to those supported by data in hand, or to explicitly state that the relevant conclusions require additional supporting data.’ Our original response letter includes a more comprehensive description of experimental revisions and was submitted in response to peer review at *Review Commons.* That original letter can be found here: https://hyp.is/go?url=https%3A%2F%2Fwww.biorxiv.org%2Fcontent%2F10.1101%2F2020.10.05.327247v1&group=NEGQVabnReviewer #1 (Evidence, reproducibility and clarity (Required)):The manuscript by Abrams and Nance describes how the polarity proteins PAR-6 and PKC-3/aPKC promote lumen extension of the unicellular excretory canal in C. elegans. Using tissue-specific depletion methods they find that CDC-42 and the RhoGEF EXC-5/FGD are required for luminal localization of PAR-6, which recruits the exocyst complex required for lumen extension. Interestingly, they show that the ortholog of the mammalian exocyst receptor, PAR-3, is dispensable for luminal membrane extension. Overall, this is a well-written and interesting manuscript.1) Because depletion of PAR-3 in the canal causes milder defects than PAR-6 or CDC-42 the authors suggest that they cannot rule out the possibility that an alternative isoform of PAR-3 is expressed and buffering the defect. They should perform canal-specific RNAi-mediated depletion of the entire PAR-3 gene to determine if this is true.

We agree that without removing the entire *par-3* gene we cannot rule out the possibility of an alternative form of PAR-3 buffering the canal defect we observe. We point out in the Results and Discussion that further experiments are needed to determine if an alternative form of PAR-3 is present.

“we cannot exclude the possibility that an undescribed isoform of *par-3* with a different 3’ end, and thus lacking the ZF1 tag, is expressed within the excretory cell and buffers mutant phenotypes”

“further experiments will be needed to determine if an alternative form of PAR-3 protein lacking the ZF1 degron is produced”

2) The authors suggest that GTP-loaded (activated) CDC-42 recruits PAR-6 to the luminal membrane. It would be nice if they could use a biosensor, such as the GBD-WSP-1 reagent from Buechner's lab to confirm that EXC-5 depletion also reduces activated CDC-42, as would be expected. This should be achievable since there is strong CDC-42 signal, even in the cytoplasm.

This is an excellent suggestion. To demonstrate this in a future report, we will utilize a CDC-42 biosensor – an integrated *cdc42p::gfp::wsp-1(gbd)* strain created in our lab and previously validated and characterized (Zilberman et al., 2017). We have added to the Discussion to highlight such future experiments.

“In addition, further experiments will be required to determine whether EXC-5 activates CDC-42 specifically at the lumenal membrane, as our model predicts, and to identify the biochemical links between EXC-5, CDC-42, PAR-6, PKC-3, and the exocyst complex.”

3) Related to point 2, (i) does mutation of the CRIB domain of PAR-6 impair its recruitment to the luminal membrane, and (ii) does this mutant exacerbate canal defects when PAR-3 is depleted?

i) Our lab has previously generated and characterized a transgenic *par6P::par-6(∆CRIB)::gfp* strain (Zilberman et al., 2017). In a future report, we will compare lumenal enrichment of PAR-6(∆CRIB)::GFP to control worms expressing wild-type PAR-6::GFP. See comments for point #2 above regarding identifying the biochemical links between CDC-42 and PAR-6.

ii) This is a very interesting experiment, as it would help address if the mild phenotype observed in PAR-3-depleted animals is due to the remaining PAR-6 that is recruited by CDC-42. Our lab has previously shown that *par6P::par-6(∆CRIB)::gfp* cannot rescue the embryonic lethality of a *par-6* mutant, in contrast to *par-6::gfp* (Zilberman et al., 2017). This indicates that the CRIB domain is needed for PAR-6 function during embryogenesis and suggests that CRIB domain mutations introduced by CRISPR would almost certainly be lethal, precluding analysis of the excretory cell. We have expanded the Discussion to acknowledge that future experiments are required to determine if CDC-42 at the lumen is required for PAR-6 recruitment as our model predicts (see point #2 above).

4) The authors hypothesize that partial recruitment of PAR-6 by CDC-42 is sufficient for luminal membrane extension to explain the mild defects caused by PAR-3 depletion. Since depletion of PAR-6 and CDC-42 alone causes milder canal truncations the authors should co-deplete these proteins (as well as PAR-3 and CDC-42) to determine if there is an additive effect.

This is an excellent suggestion in principle. However, it is not possible to know in any given degradation experiment whether the targeted protein is completely degraded; we can only say it is no longer detectable by fluorescence. Thus, any degron allele (in the presence of ZIF-1) could behave like a strong hypomorph rather than a null. It would not be possible to interpret double degradation experiments in such a case, as a more severe phenotype in the double could simply be a result of combining two hypomorphic alleles, further reducing pathway activity even if the genes function together. To interpret this experiment properly, a null allele of at least one of the genes would have to be used. This is not possible since *par* and *cdc-42* null mutants are lethal and there is also maternal contribution. We have added to the Results and Discussion to acknowledge the possibility that the degron alleles may not represent a null phenotype, perhaps due to variable protein degradation rates in each ZF1 allele. These subtle differences in ZF1 mediated degradation could explain some of the milder phenotypes in PAR-6 or CDC-42 depleted larvae.

“Whereas the phenotype of PAR-3^exc(-)^ larvae appears distinct, more subtle differences in excretory canal length following the depletion of specific proteins might reflect variation in degradation rates or efficiency (Nance and Frokjaer-Jensen, 2019).”

“Finally, it is possible that degradation of the targeted ZF1-tagged proteins, while visibly below our level of detection by fluorescence, is not complete and phenotypes are hypomorphic.”

5) In Figure 2, the authors show that depletion of PKC-3 causes more severe canal truncations than PAR-6. Since these proteins function in the same complex what do they think is the reason for this difference? This point could be discussed more in the manuscript.

As described in the previous point, incomplete degradation could produce modestly different phenotypes even for genes that act in the same pathway. Therefore, it is not possible to determine whether PAR-6 and PKC-3 have different roles using this approach. We have added text to the Discussion to clarify that we cannot conclude from our current findings that PAR-6 and PKC-3 have equivalent roles in the excretory canal.

“Although PAR-6 and PKC-3 bind one another and are typically thought to function as an obligate pair, we note that our experiments do not directly address whether they function together in lumen extension.”

6) Related to point 5, more experiments with PKC-3 should be done to determine if, for example, localization of S-10 is similarly affected as ablation of PAR-3, PAR-6 and CDC-42.

We agree that our current results do not directly demonstrate that PKC-3 is necessarily acting similarly to PAR-6 in the excretory cell. We have added to the Discussion to clarify that while it is likely that PAR-6 and PKC-3 have a similar function in the excretory canal, additional experiments will be needed to determine this. (see point #5)

Reviewer #2 (Evidence, reproducibility and clarity (Required)):The manuscript by Abrams and Nance describes a precise investigation of the role of PAR proteins in the recruitment of the exocyst during and after the extension of the C. elegans excretory canal. State-of-the-art genetic techniques are used to acutely deplete proteins only in the targeted cell, and examine the localization of endogenously expressed markers. Experiments are well described and carefully quantified, with systematic statistical analysis. The manuscript is easy to follow and the bibliography is very good. Most conclusions are well supported.1) I am not entirely convinced by the presence of CDC-42 at the lumenal membrane (Figure 3G); it seems to be more sub-lumenal that really lumenal. It peaks well before PAR-6 (Figure 3H) which itself seem slightly less apical that PAR-3 (Figure 3F). Could you use super-resolution microscopy (compatible with endogenous expression levels) to more precisely localize CDC-42? Similar point for PAR-3 and PAR-6 which do not seem to colocalize completely – a longitudinal line scan along the lumenal membrane might provide the answer even without super-resolution; this could help explain why these two proteins do not have the same function. These suggestions are easy to do provided the authors can have access to super-resolution (Airyscan to name it; although other methods will be perfectly acceptable I believe it is the most simple one).

We agree that the CDC-42 localization peak does not precisely match the PAR-6 peak. As the reviewer notes, resolving the subcellular localization of these two proteins will not be feasible using standard confocal microscopy. While the peak localization intensities do not coincide, like they do for PAR-6 and PAR-3, super-resolution imaging will be required to determine whether CDC-42 is present or excluded from the lumenal domain. Higher resolution imaging will also help resolve if PAR-3 and PAR-6 are expressed in distinct puncta along the lumenal membrane, which could explain their distinct functions as the reviewer points out.

We have added text to the Results to clarify that we will need super-resolution imaging to determine if CDC-42 is present at the lumenal membrane and to further resolve the colocalization of PAR-3 and PAR-6.

“While the peak localization intensities of ZF1::YFP::CDC-42 and PAR-6::mKate in transects across the width of the excretory cell do not align, as they do with PAR-6::ZF1::YFP and PAR-3::mCherry, super-resolution imaging would be required to determine whether ZF1::YFP::CDC-42 is present at the lumenal domain.”

2) The same group has described a CDC-42 biosensor to detect its active form. It could be used here to precisely pinpoint where active CDC-42 is required: in the cytoplasm? At the lumenal membrane? colocalizing with what other protein? This will require the expression of a transgene under an excretory cell specific promotor and a simple injection strategy while helping to strengthen the description of the CDC-42 role.

See reviewer 1 point #2.

3) As the authors certainly know, there is a PAR-6 mutation which prevents its binding to CDC-42. They could express this construct in the excretory canal a simple extrachromosomal array should be sufficient) to validate the direct interaction between these proteins in this cell.

See reviewer 1 point #3.

4) What is the lethality of ZIF-1-mediated depletion of the various factors under the exc promoter? Can homozygous strains be maintained? Authors just have to add a sentence in the Materials and methods section.

All of the strains with excretory cell-specific degradation we have examined are viable when grown on NGM plates. We have added this point to the Materials and methods.

“SEC-5^exc(-)^, RAL-1^exc(-)^, PKC-3^exc(-)^, PAR-6^exc(-)^, CDC-42^exc(-)^, and PAR-3^exc(-)^ strains were all homozygous viable when grown on NGM plates.”

*Provided that the authors have access to an Airyscan, all the questions asked here can be answered in two months (one month for constructs, one month for injection and data analysis) at a very minor cost*.

Reviewer #3 (Evidence, reproducibility and clarity (Required)):Strengths of this manuscript include the use of endogenously tagged proteins (rather than over-expressed transgenes) for high resolution imaging and a cell-type specific acute depletion strategy that avoids complicating pleiotropies and allows tests of molecular epistasis. While some results were fairly expected based on prior studies of Cdc42, PAR proteins, and the exocyst in other tissues or systems, differences in the requirements for par-6 and pkc-3 vs. par-3 strongly suggest that the former genes play more important roles in exocyst recruitment. I was also excited to see a connection made between EXC-5 and PKC-3 localization.1) Lumen formation vs. lumen extension. The Abstract and Introduction use these two terms almost interchangeably, but they are not the same and more care should be taken to avoid the former term. The data here do not demonstrate any roles for par or other genes in lumen formation, but do demonstrate roles in lumen extension and organization/shaping.

We agree and have corrected wording throughout the manuscript to indicate that lumen *extension* is affected.

2) Related to the above, mutant phenotypes here are surprisingly mild and variable. The authors discuss possible reasons for the particularly mild phenotype of par-3 mutants, but don't specifically address the mild phenotypes of the others. Clearly quite a bit of polarization and apical membrane addition occurs in ALL of the mutants. Is this because those early steps use other/redundant molecular players, or is depletion too late or incomplete to reveal an early role?

We agree with reviewer 3 and we have added these points in the Discussion. Degradation of proteins strongly predicted to function together (RAL-1 and SEC-5; PAR-6 and PKC-3) produce similar although not identical phenotypes; as discussed above we consider it likely that these differences reflect minor differences in degradation efficiency below our ability to detect by fluorescence. As reviewer 3 points out, the excretory-specific driver we use to express ZIF-1 may not be active at the very earliest stages of lumen formation, and degradation could take 45 minutes or more after the promoter becomes active (Armenti et al., 2014). Thus, we agree that phenotypes could be more severe if it were possible to completely deplete each tagged protein prior to the onset of lumen formation. However, this caveat does not change the interpretations of our experiments since all proteins are degraded with the same driver. We have avoided mentioning that the phenotypes we observe reflect the “null” phenotype for these reasons. We have now emphasized these points in the Results and Discussion.

“Whereas the phenotype of PAR-3^exc(-)^ larvae appears distinct, more subtle differences in excretory canal length following the depletion of specific proteins might reflect variation in degradation rates or efficiency (Nance and Frokjaer-Jensen, 2019).”

“Even though lumen extension is severely compromised in SEC-5^exc(-)^, RAL-1^exc(-)^, PAR-6^exc(-)^, PKC-3^exc(-)^, and CDC-42^exc(-)^ larvae, the initial stages of lumenogenesis still occur. One possible explanation is that a distinct pathway directs the initial stages of lumen formation. Alternatively, since it is unclear whether the *excP::zif-1* transgene is active at the very early stages of lumenogenesis (see Results), it is possible that complete loss of the targeted proteins immediately after excretory cell birth would block lumen formation entirely. Finally, it is possible that degradation of the targeted ZF1-tagged proteins, while visibly below our level of detection by fluorescence, is not complete and phenotypes are hypomorphic. Resolving these possibilities will require the use of earlier-acting *zif-1* drivers and alternative genetic methods.”

The authors introduce a new reagent, "excP" (the promoter for T28H11.8), which they use to drive canal cell expression of ZIF-1 for their degron experiments. Please provide more information about when in embryogenesis this promoter becomes active, how that compares to when the par genes, sec-5, ral-1 and cdc-42 are first expressed, and what canal length is at that time. It would also be helpful to show the timeframe for degron-based depletion using this reagent (Figure 1C shows only depletion at L4, days later).Publicly available single cell RNA seq data (https://pubmed.ncbi.nlm.nih.gov/31488706/ and https://cello.shinyapps.io/celegans_explorer/) suggest that canal expression of the endogenous T28H11.8 gene doesn't really ramp up until the 580-650 minute timepoint, which is several hours after par gene canal expression (270-390 minutes) and the initiation of canal lumen formation (bean stage, 400-450 minutes). These data suggest that excP might come on too late to test requirements in lumen formation and early stages of extension. This caveat should be at least mentioned.

See point #2 above. We agree that providing more information on expression from the *T28H11.8* promoter would be important for interpreting the severity of phenotypes. Since we have not measured what level of *T28H11.8* expression is needed to produce a sufficient amount of ZIF-1 for degradation, we have refrained from estimating a specific timeframe of degron-based depletion, but we have added to the text the caveats to this method (see point #2). We have also referenced the *T28H11.8* single cell RNA seq data in the Results – thank you to the reviewer for this suggestion.

“endogenous *T28H11.8* mRNA is first detected by single-cell RNA sequencing in the excretory cell several hours after its birth (Packer et al., 2019).”

3) There are two major aspects to the mutant phenotypes observed here: short lumens and cystic lumens. A short lumen makes sense intuitively, but the cysts could use a little more explanation. (What are cysts? What is thought to be the basis of their formation?). It is intriguing that cysts in sec-5 vs. ral-1 mutants (Figure 1) and par-6 vs. pkc-3 mutants (Figure 4) seem to have a very different size and overall appearance. Are these consistent differences, and if so, what could be the explanation for them?

This is an interesting point. Since it is not practical to perform time-lapse imaging to watch canal cysts form, we analyzed only L1 and L4 larvae. We believe from our imaging that these are discontinuous regions of the lumen. One explanation for the expansion and dilation of the cystic lumens by L4 stage could be that the canal lumen has been expanded by fluid buildup resulting from a defect in canal function in osmoregulation, but we have not tested this directly. The reviewer also raises an interesting point regarding different appearances of cysts in SEC-5 and RAL-1 depleted larvae compared to PAR-6 and PKC-3. It is possible that these differences arise because SEC-5 and RAL-1 direct whether vesicles will fuse at all, whereas PAR proteins direct where they will fuse in the cell. We have added a description of the cystic lumens to the Results.

“Small cysts often appeared to be discontinuous, although given the resolution of our imaging, it is possible that they remain connected by small bridges. In addition, we note that the size of cysts could be affected by swelling of the lumen as an indirect consequence of poor osmoregulation.”

4) The authors did not test if PKC-3, like PAR-6, is required to recruit exocyst to the canal cell apical membrane, but their prior studies in the embryo suggested that it is (Armenti et al., 2014). They also did not test if EXC-5 is required to recruit PAR-6 and the exocyst (along with PKC-3), or if CDC-42 is required to recruit PKC-3 (along with PAR-6). There seems to be an assumption that PAR-6 and PKC-3 are regulated and function in a common manner (as is often the case), but that has not been demonstrated here specifically. The basis for this assumption and alternatives to the linear model should be acknowledged.

A related point was raised by reviewer 1 (see reviewer 1 point #6). We agree with reviewer 3 that we have not shown that PAR-6 and PKC-3 always function similarly, although this is expected based on their similar phenotypes and co-dependent functions in other cells. We have added this caveat in the Discussion.

“Although PAR-6 and PKC-3 bind one another and are typically thought to function as an obligate pair, we note that our experiments do not directly address whether they function together in lumen extension.”

5) EXC-5 is presumed to act upstream of CDC-42 based on shared phenotypes and the known Rho GEF activity of its mammalian homologs. However, direct evidence for this is currently lacking. In future, the authors might test if depleting EXC-5 affects CDC-42 activation/GTP-loading by using CDC-42 biosensors that have been reported in the literature (e.g. Lazetic et al., 2018).

See reviewer 1 point #2.

Minor comments:Figure 1, Figure 4, Figure 4—figure supplement 1 and 2Blue color/CFP indicates the apical/luminal membrane or the apical region of the canal cytoplasm, not the actual lumen as the labels suggest. The lumen is a hollow cavity on the opposite side of the plasma membrane from these markers, and it is shown as white in the Figure 1A upper right cartoon.

Thank you for pointing this out. We have corrected the figure labelling.

Figure 2, Figure 2—figure supplement 1I'm not confident in the statistical analysis used here (Fisher's Exact test on two bins, <50% and >50% canal length), given that four length bins (not two) were defined. I recommend consulting a statistician.

Our rationale for using two bins for the statistical analysis was because control larvae nearly all have a similar canal length (L1 stage: 99% of larvae have canal length that is 51-75% of body length; L4 stage: 98% of larvae have canal length that is 76-100% of body length), making it straightforward to ask if mutants are shorter. We chose not to make more granular phenotypic comparisons, as we cannot rule out that subtle differences in degradation efficiency, rather than differences in biological function, underlie any differences in canal length of the degron mutants (see point #2).

In consulting biostatistics references, we concluded that small numbers of larvae (<10) within individual bins could cause the *P* value to be inaccurate for a test of independence; we have several bins in which there were small numbers of larvae in our dataset. In such an instance it is recommended to pool categories that contain small numbers (McDonald, 2014. Handbook of Biological Statistics. Vol. 3rd ed. Sparky House Publishing, Baltimore, Maryland. p. 86-89).

We have added these points to the Materials and methods.

"Born during late embryogenesis…" Actually, the canal cell is born at ~270 minutes after first cleavage, which is in the first half of embryogenesis, not what I would call "late".

We agree and have corrected the wording.